# Comprehensive Insights into Artificial Intelligence for Dental Lesion Detection: A Systematic Review

**DOI:** 10.3390/diagnostics14232768

**Published:** 2024-12-09

**Authors:** Kubra Demir, Ozlem Sokmen, Isil Karabey Aksakalli, Kubra Torenek-Agirman

**Affiliations:** 1Department of Computer Engineering, Erzurum Technical University, 25040 Erzurum, Türkiye; kubra.demir90@erzurum.edu.tr; 2Department of Industrial Engineering, Erzurum Technical University, 25040 Erzurum, Türkiye; ozlem.sokmen@erzurum.edu.tr; 3Department of Dentomaxillofacial Radiology, Ataturk University, 25240 Erzurum, Türkiye; ktorenek@gmail.com

**Keywords:** dental lesion detection, systematic review, artificial intelligence, challenges, proposed solutions

## Abstract

**Background/Objectives:** The growing demand for artificial intelligence (AI) in healthcare is driven by the need for more robust and automated diagnostic systems. These methods not only provide accurate diagnoses but also promise to enhance operational efficiency and optimize resource utilization in clinical workflows. In the field of dental lesion detection, the application of deep learning models to various imaging techniques has gained significant prominence. This study presents a comprehensive systematic review of the utilization of deep learning methods for detecting dental lesions across different imaging modalities, including panoramic imaging, periapical radiographs, and cone-beam computed tomography (CBCT). A systematic search was conducted following the Preferred Reporting Items for Systematic Reviews and Meta-Analyses (PRISMA) guidelines to ensure a structured and transparent review process. **Methods:** This study addresses four key research questions related to the types of objects used for AI in dental images, state-of-the-art approaches for detecting lesions in dental images, data augmentation methods, and challenges and possible solutions to the existing AI-based dental lesion detection. Furthermore, this systematic review was performed on 29 primary studies identified from multiple electronic databases. This review focused on studies published between 2019 and 2024, sourced from IEEE, Web of Knowledge, Springer, ScienceDirect, PubMed, and Google Scholar. **Results:** We identified five types of lesions in dental images as periapical lesions, cyst lesions, jawbone lesions, dental caries, and apical lesions. Among the fourteen state-of-the-art deep learning approaches, the results demonstrate that deep learning models, such as U-Net, AlexNet, and You Only Look Once (YOLO) version 8 (YOLOv8) are commonly employed for dental lesion detection. These deep learning models have the potential to serve as integral components of decision-making processes by improving detection accuracy and supporting clinical workflows. Furthermore, we found that among twelve types of data augmentation techniques, flipping, rotation, and reflection methods played an important role in increasing the diversity of the datasets. We also identified six challenges for dental lesion detection, and the main issues were identified as data integration, poor data quality, limited model generalization, and overfitting. Proposed solutions against the aforementioned challenges include the integration of larger datasets, model optimization, and diversification of data sources. **Conclusions:** This study provides a comprehensive overview of current methodologies and potential advancements in dental lesion detection using deep learning. The findings indicate that possible solutions against the challenges of AI-based diagnostic methods in dental lesion detection need to be more generalizable regardless of image type, the number of data, and data quality.

## 1. Introduction

Dental lesions may arise from both odontogenic and nonodontogenic origins. In radiography, depending on the density of the nearby bone, they can be characterized as either radiolucent, radiopaque, or mixed in appearance [1]. In the asymptomatic cases, pathologies can be detected on panoramic and periapical radiographs taken during routine examination. But sometimes, diagnosis may be challenging because both odontogenic and nonodontogenic lesions may mimic each other with similar radiological appearances or the lesions may be overlooked due to distortion, magnification, and superpositions seen on radiographs. In such cases, cone-beam computed tomography (CBCT), an advanced imaging method, is frequently used in dentistry. It provides ease of diagnosis for physicians thanks to its three-dimensional and non-superposition imaging features. However, it causes cost, time, and labor loss and an additional radiation dose [2]. Some imaging techniques are seen in Figure 1.

Integrating artificial intelligence (AI) usage in healthcare is promising to reduce physician workload and prevent oversights, particularly with the successful results of deep learning models. Studies in this field have demonstrated that the use of different AI models can produce different results depending on the conditions such as the created dataset, disease type, and imaging techniques [3]. Several factors affect the accuracy of the models. A deep learning model is expected to detect dental diseases with as high accuracy as possible without the need for a dentist’s control. In the literature, there are studies in which lesion detection is performed using different imaging techniques and deep learning models. Considering the existing studies, many lesion detection method experiments were carried out by creating different imaging techniques and conditions. The results obtained vary depending on variables such as the selected deep learning model, imaging technique, dataset used, opinion from experts, and lesion type. Considering these and similar situations, conducting a systematic review of existing studies is important before starting to work in the relevant field. Various research questions were identified for the use of deep learning in lesion detection within the scope of the systematic review. The primary objective of this study is to provide valuable insights that can guide and inform future research in this field. As a result of the answers given to the questions, the types of dental lesions, the most commonly used deep learning models for detection, methods that can be used in data augmentation, and effective solutions to the difficulties encountered during these processes are found.

Within the scope of this study, various research questions were identified for the use of deep learning in lesion detection. Evaluating the studies according to the external and internal criteria determined within the scope of the research sought answers to these questions. This systematic review addresses detection challenges related to lesion types, state-of-the-art approaches, data augmentation methods to improve detection accuracy, and challenges and research gaps in this field. Additionally, the integration of data-driven solutions, such as deep learning models, has the potential to optimize clinical workflows and resource utilization within the healthcare sector. This can be achieved by improving detection accuracy, reducing time spent on diagnosis, and enhancing overall decision-making processes. In the following sections of this paper, we describe the research objectives and methodology of this systematic review, including the goals and research questions, as well as the processes for data extraction, synthesis, and reporting (Section 2). Then, the selection process of the primary studies and the findings are given in the Results Section (Section 3), and finally, the results are handled with the Related Work, Limitations and Potential Threats to Validity, and Conclusions subsections in the Discussion Section (Section 4).

## 2. Methods

Adhering to Preferred Reporting Items for Systematic Reviews and Meta-Analyses (PRISMA) [4] guidelines, this study was registered with the International Prospective Register of Systematic Reviews (PROSPERO) under registration number CRD42024607099.

### 2.1. Research Objectives and Methodology

This section describes the research objectives and methods used in this study during the systematic review phase. Our research methodology was established based on the guidelines suggested by Kitchenham et al. [5], Wohlin [6], and PRISMA updated guidelines [4]. Three main activities are carried out in this section: (1) describing the purpose of this study and research questions, (2) extracting data using the data extraction form, and (3) synthesizing and reporting the data. The details of each step are explained below under subheadings.

#### 2.1.1. Goal and Research Questions

The scope and purpose of this study were determined based on the Goal–Question–Metric approach proposed by Basili [7] as follows:**Analyze** the lesion detection and segmentation using deep learning models in dental images;**For the purpose of** identification and analysis;**With respect to** object types in dental panoramic/periapical/CBCT images, state-of-the-art solutions, data augmentation methods to detect lesions in dental images, possible research directions, and future developments;**From the point of view of** dentists and computer science researchers;**In the context of** deep learning.

This study aimed to systematically classify, examine, and synthesize the body of knowledge and evidence regarding the use of deep learning algorithms for lesion detection in dental images through a systematic review. As stated by Kitchenham et al. [8], it is believed that the findings obtained as a result of the reviews conducted on the following research questions will pave the way for secondary studies.


*RQ1. What are the object types for detection and classification in dental panoramic/periapical/CBCT images?*
This research question aims to determine the application areas of AI in dental images by investigating what types of objects can be detected in different dental imaging techniques in the literature.


*RQ2. What are the state-of-the-art approaches to detect lesions in dental images?*
This research question focuses on the analysis of current AI-based solutions applied by dentists to detect lesions that are relatively difficult to distinguish by the eye compared to other objects.


*RQ3. What are the data augmentation methods used for dental images?*
This research question aims to investigate the data augmentation methods applied due to the difficulty of object detection in dental images and the effects of these methods on model performance.


*RQ4. What are the challenges and proposed solutions in dental lesion detection?*
This research question aims to guide future researchers by including the difficulties encountered in AI-based dental lesion detection and solutions to these difficulties.

#### 2.1.2. Data Extraction

The data extraction form consists of search queries, research questions, search engines, study selection process, inclusion–exclusion criteria, quality metrics, and findings related to the research questions obtained from the primary studies. The search queries shown in Table 1 are lesion detection OR object types AND dental images; lesion detection AND deep learning methods; state-of-the-art solutions AND dental images; and application areas AND lesion detection. The first six lines of this form capture the metadata of the articles. We defined several parameters and used inclusion and exclusion criteria to extract data from the selected studies, thus facilitating data management. The quality criteria are kept in the same format as the parameters used for the research questions, as they are crucial for data analysis. The studies in the form include fields for study ID, title, authors, year, publication type, publisher, and additional notes. Furthermore, the final results from the primary studies for each research question are categorized in this form. The data extraction form is provided as Appendix A.

After determining the keywords, primary studies were obtained from the analyzed articles. The inclusion and exclusion criteria applied for the study selection process are given in Table 2, respectively.

#### 2.1.3. Data Synthesis and Reporting

During the data synthesis and reporting process, quantitative and qualitative analyses of all research questions were conducted and percentage information and categorical data were provided regarding which primary study the findings obtained from each research question were included in. During data extraction, qualitative data such as object types, state-of-the-art solutions, and data augmentation methods obtained from research questions were recorded in the data extraction form created as a Google Sheet. This form included a table indicating whether each study included the obtained items.

## 3. Results

### 3.1. Study Selection

In the primary study selection process, six databases containing many articles in the fields of health and computer sciences, namely, IEEE Explore, Web of Science, Springer, Google Scholar, Science Direct, and Pubmed, were used. To focus on recent studies in particular, studies obtained between 2018 and 2024, which appeared on the first two pages of the databases, were included in the study selection process, and a total of 350 studies were considered. Table 3 shows the total number of studies per database and the number of selected primary studies according to inclusion and exclusion criteria. Among these 350 studies, 29 studies were primarily selected according to the inclusion and exclusion criteria given in Table 4. The flow diagram of the primary study selection is presented in Figure 2. The protocol of this review was developed according to the PRISMA statement [9,10]. The publication channels of the selected primary studies are given in Table 5, respectively.

The PRISMA flow diagram illustrates the selection process applied to a dataset comprising 350 studies. Following the application of inclusion and exclusion criteria, 37 duplicate studies were excluded. The remaining 313 studies underwent detailed evaluation, resulting in the exclusion of 284 studies. Consequently, 29 studies were deemed eligible and included in this review. This process underscores the rigorous evaluation undertaken to ensure that only studies meeting the predefined criteria were selected.

In the final stage of the selection process of primary studies, the quality criteria of the studies were scored as “yes = 1”, “somewhat = 0.5”, and “no = 0” according to the quality assessment criteria suggested by Kitchenham et al. [40]. The selection of the high-quality primary studies was performed according to the average score exceeding 5. As a result of the quality assessment, it was seen that the studies had at least 5 points and therefore it was decided to evaluate all the primary studies selected according to the inclusion/exclusion criteria. The questions suggested by Kitchenham et al. [40] for quality assessment are given in Table 6.

Some of the articles that are systematically reviewed in the literature are considered primary studies. It is a prerequisite for determining primary studies that meet both external and internal criteria in accordance with the keywords. If even one of the external criteria can be selected, that study is not primary. As shown in Figure 3, when examining the distribution of studies by year, it can be observed that the detection of dental lesions using deep learning methods was least frequent in 2019 based on primary studies. It can be observed that the number of primary studies was particularly high in 2022. These studies indicate that the detection of dental lesions is becoming increasingly important. The vast majority of the studies conducted have been published in journals. Publication types such as conferences come after article studies. In the continuation of the determined primary studies, answers are sought according to the questions. The quality criteria of the answers are determined. According to these criteria, it is learned whether the study provides sufficient information. Research methods were carried out on experimental studies based on lesion detection.

### 3.2. Quality Assessment

The studies are divided into four parts when they are evaluated. These are determined as reporting, rigor, credibility, and relevance. In the reporting quality control, the purpose, scope, and context are examined and an evaluation is made as to whether it is an experimental study. In the rigor section, three different questions are examined as to whether the study is a valid and reliable source, whether sufficient information is documented, and whether the questions are answered. Then, the reliability of the study is evaluated according to the mentioning of negative findings in the article and the reputation of the article. In the last section, the relevance stage, the quality control decision is made according to the finalization of the content according to the purpose and whether the research is good in terms of practicality. From here, good, medium, and bad interpretations are made according to the metrics given between 1 and 10. Good corresponds to 1.0, medium to 0.5, and bad to 0.0. Accordingly, it was possible to see which questions were answered in the rigor quality criterion. As given in Table 7, the quality assessment is conducted according to the criteria.

According to the graph given in Figure 4, it is seen that there are 7 studies that received 9.5 and 9.0 total quality scores out of 29 primary studies. When the evaluation is examined, it is seen that there are three studies with 7.5, 7.0, and 6.5 scores. Apart from these, there are five studies with 8.5 and one study with 8.0. The figure shows that no quality score of 6.0 or below was received. This means that the primary studies are largely in compliance with the determined quality criteria.

When the graphs obtained from the quality assessment criteria are examined individually, it is observed that some are evaluated based on three metrics, while others are assessed using two metrics. The reporting criterion, presented in Figure 5, includes three metrics. It is noted that 18 studies, which include all three metrics, achieved a score of 3.0 among the primary studies. Eight studies received a score of 2.5, while three studies scored 2.0. The absence of any studies with a score below 2.0 indicates that the selected primary studies achieved satisfactory outcomes in terms of purpose, scope, context, and experimental study criteria.

The rigor values presented in Figure 6 are an effective criterion, particularly for addressing questions in primary studies. It has been observed that the articles received scores ranging from a maximum of 3.0 to a minimum of 1.0. Among the studies, three achieved a score of 3.0, while the twelve articles with the highest frequency received a score of 2.5. Nine studies ranked second, scored 2.0, and were selected from the primary studies. Among the remaining five studies, three received a score of 1.5 and two received a score of 1.0. It can be concluded that the 0.5 point deduction in scores for individual questions is primarily due to the limited number of studies that provided complete answers to all four questions. Similar to the external criteria used in this study, articles missing even one of the four questions were classified as medium-level with a score of 0.5.

In addition to the experimental results of primary studies, reliability is also a critical criterion. For this purpose, the reliability measure was used. This measure examined the negative findings reported in the documents and the reputation of the studies. The negative findings highlighted in the studies have a significant impact on future research and therefore play an important role in reliability. Similarly, the reputation and validity of the studies are important factors in the selection of primary studies. Among the 29 articles selected for the systematic review, 12 articles received a score of 1.5, exceeding the maximum score of 2.0 in terms of negative findings and study reputation. As seen in Figure 7, 11 studies received a score of 2.0, 5 studies received a score of 1.0, and 1 study received the lowest score of 0.5.

The main purpose of the last criterion, relevance, is to examine the article in general. It focuses on the results of the findings obtained. In this context, two metrics were determined. In the first of these, the articles were examined by searching for answers to the questions “Is the result connected to the purpose?” and in the second, “Has enough research and practice been undertaken for the study?” When Figure 8 is examined, 23 articles were evaluated with a maximum score of 1.0 out of 29 articles within the scope of these two metrics. Here, six articles were found with a score of 0.5. No article reaching a score of 2.0 in terms of relevance could be obtained. However, it can be said that this situation is mostly due to the lack of sufficient development in the research and practice sections.

Quality control performed based on criteria is important in terms of finding the most accurate answers to questions from primary studies. Therefore, by using the obtained quality score values, studies that need to be focused on for answers to questions from articles are revealed. After determining the priority studies, the answers to the questions can be obtained automatically. Reporting, rigor, credibility, and relevance criteria play a decisive role in this regard.

### 3.3. Answers to Research Questions

In this section, primary studies are examined for the answers to the questions. In finding the answer to the first question, object types that can be detected through panoramic, periapical, and CBCT image types are sought. These types were determined as lesion-based periapical, cyst, jawbone, tooth decay, and apical in this study. In the second question, primary studies were examined in terms of approaches that can be used in the detection of these objects. What is mentioned here is the determination of deep learning methods used in the studies. The third question explored data augmentation methods employed for dental images, identifying the most effective approaches. In the last research question, we investigated challenges and possible solutions to these challenges in dental lesion detection. The answers to these questions are presented in detail through summary tables in this study.

#### 3.3.1. What Are the Object Types for Detection and Classification in Dental Panoramic/Periapical/CBCT Images?

Various imaging techniques have been utilized for dental images over time. The primary objective of each new technique is to enhance the ability to distinguish objects in the image with greater precision. In this systematic review, various types of lesions identified by dental panoramic, periapical, and CBCT imaging techniques were discussed in the articles, and the performances of the models used were examined. Upon examining the selected primary articles, it was observed that performance evaluation was typically conducted using a single technique in some studies, while others employed multiple imaging techniques. Considering this situation, the lesion types that can be obtained were periapical, cyst, jawbone, tooth decay, and apical, and these objects were determined as lesions. These types are explained in detail in this section, and the distribution of object types considered according to primary studies is given in Table 8.

*Periapical lesion:* One of the most prevalent dental diseases is periapical lesions. In order to assess common clues about the diagnosis of periapical lesions, clinical and radiographic examinations are essential [27]. Periapical radiographs, panoramic radiographs, and CBCT are imaging methods used in radiographic examination to assess the presence of periapical lesions [41]. However, the frequency of detection decreases for panoramic and CBCT images, most of which also involve periapical lesions. S1 is numbered as the first primary study identified in the Springer database during the search. In the study S1 [11], the evaluation of lesions in the roots of teeth was performed using deep learning on periapical radiographs. Within the scope of the study, 3000 periapical root areas were extracted from 1950 images, and scoring was performed according to the size of the lesion in the root on the resulting data. Then, periapical lesions were graded, and according to this grade, it was compared whether the model used with different metrics gave good results. As a result of the study, it was observed that some of the root scores such as 1, 2, 3, 4, and 5 where periapical lesions were graded gave good results, while some scores did not give good enough results. Therefore, it was thought that the model needed to be developed and needed more training. In the remaining 17 studies, different imaging techniques were performed based on the periapical lesion object. 

*Cyst lesion:* The most common bone in the human body to develop cysts are the jaw, which, due to growing dentition, is closely related to the several epithelial rests. Because of their similar clinicopathological and radiographic presentations, many cysts in the jaws might mimic tumors and intraosseous lesions [42]. Many deep learning-based studies have been carried out to detect these lesions. The S2 study [12] aimed to detect cyst-like lesions. In the study, panoramic dental X-Rays of 412 patients were examined, and maxillary cyst-like lesions were detected in these X-Rays. As a result of the study, it was stated that deep learning methods could achieve effective results in lesion detection, but further research was required for this. Similarly, the study S25 [35] addressed the confusion between radicular cysts and periapical lesions. On panoramic imaging, dentists frequently struggle to differentiate radicular cysts from periapical granulomas. Root canal therapy is the initial line of treatment for periapical granulomas, whereas surgical removal is necessary for radicular cysts. Thus, there is a need for an automated tool to support clinical decision-making [35]. The deep learning method was applied on panoramic radiographs of teeth with 80 radicular cysts and 72 periapical lesions previously determined in the dataset. When the obtained results were evaluated, it was thought that more research was needed for periapical lesion detection within the scope of the study and that automatic detection could be achieved by trying the deep learning method. In these two studies, in which cyst and periapical lesion object types were detected, it was seen that the second study achieved better results compared to the first when the cyst was focused on. 

*Jawbone lesion:* The most common pathologies encountered in the jawbones are cysts, tumors, and tumor-like diseases. Since the identification of lesions on panoramic radiography has significant clinical importance, intensive efforts have been made to develop deep learning-based models for pathological diagnosis. Despite these advances, the search for a method that can be effectively used in the clinic to diagnose jawbone pathologies continues [43]. The study S3 [13] focused on the detection of jawbone lesions based on deep learning methods. Following the implemented operations, it was observed that the diagnostic accuracy could reach up to 96.57% when 5% of the dataset—labeled with expert input and containing sufficient data—was utilized. The careful use of expert opinion in the labeling process increased sensitivity and specificity as well as accuracy. The study shows that when deep learning and carefully labeled data are used, detection from medical images can achieve highly accurate results. Based on this, the detection of the jawbone object was provided by the deep learning method. 

*Tooth decay lesion:* Tooth decay is a pathology that starts from the tooth enamel and causes tooth loss, especially as it progresses, and is considered the beginning of dental diseases. In addition, it can progress to the formation of cysts and tumor-like structures at later levels. Based on this, it was considered as an object type that required detection and classification for the lesion in the study [44]. In this context, studies S5 and S6 from the IEEE database were considered as primary studies, and the detection of the dental caries object was performed based on deep learning. In the study S5 [15], a new deep learning method was developed, and automatic detection of dental caries lesions was achieved. In the study conducted on periapical images, the sensitivity of caries lesions resulted in a high value of 99.13%. It was determined that the developed model was effective in detecting dental caries lesions. In the other article, S6 [16], the authors searched for methods to facilitate the treatment process of common dental diseases such as dental caries and missing teeth using AI and image techniques. Today, dentists’ manual search for lesions by eye both is a waste of time and sometimes leads to incorrect results. In the study in question, firstly, AI was used to detect dental diseases such as lesions, dental caries, and missing teeth using learning transfer methods. Although the target here was not exactly caries, dental diseases were detected in general. 

*Apical lesion:* The term apical periodontitis, which is the beginning of periapical tissue diseases, is generally used to explain the onset of various periapical conditions originating from pulp diseases, which are named and grouped according to the developmental stages of the disease [45]. Although apical and periapical lesions are combined in the same sense, it was deemed appropriate to present them under two separate headings as used in the articles included in this study. When the content of this study is examined, it is seen that apical lesion detection was performed in 10 articles. Among these, the three selected studies were S7, S8, S9, respectively. In the primary study S7 [17], the authors aimed to detect lesions in tooth roots, and the detection process was carried out using different deep learning methods on 660 images. After these processes, the results were compared and the method with the highest accuracy was decided. In the primary study S8 [18], the authors argue that different imaging techniques can be used in addition to the X-Ray images that are mostly used in the detection of apical lesions and that it may have a more facilitating effect on lesions detected manually. The imaging techniques discussed here are panoramic, periapical, and CBCT radiographs. In the scope of the study, a database was created for these images, and a new Convolutional Neural Network (CNN) proposal was given. Within the scope of the study S9 [19], the detection of lesions on periapical images was conducted. The purpose of this detection was to reduce the workload of dentists and save them from the difficulty of manual labeling. For this purpose, an analysis method was suggested within the scope of the study. Data were obtained through a database created by expert dentists. The detection process was performed using a neural network. The accuracy rate increased by more than 5% compared to the currently available methods. As a result, both time and treatment were saved. When these studies were examined, the apical lesion was also accepted as an object type that required detection in this study, presented as a separate group.

#### 3.3.2. What Are the State-of-the-Art Approaches to Detect Lesions in Dental Images?

Certain technological approaches are required for vital operations such as the detection and classification of dental diseases. Considering this situation, the second question is to examine what the latest technological approaches can be in detecting lesions in dental radiographic images. The latest approaches for detecting lesions in dental images can be used for different situations. The main deep learning models found in primary studies are U-Net, AlexNet, You Only Look Once (YOLO) version 3 (YOLOv3), YOLOv5, YOLOv8, CNN, GoogleNet, Denti.Al, Visual Geometry Group (VGG16, VGG19), DentaVN, RetinaNet, SqueezeNet, Segment Anything Model (SAM), ResNet50, DetectNet, and MobileNetV2. These approaches perform classification, segmentation, and detection processes. Classification seeks to determine whether an image belongs to a specific class. Detection involves identifying specific objects within an image and annotating their locations. Segmentation, in contrast, performs a comprehensive pixel-level analysis to assign each pixel in the image to a particular object or class. Some approaches can perform these operations together. CNN, GoogleNet, and MobileNetV2 approaches classify whether there is a lesion or not. Segmentation and classification are carried out concurrently in CNNs. Additionally, segmentation is a core component of the U-Net architecture. In this study, the approaches and operations employed differ depending on the specific context, as detailed in Table 9.

*U-Net:* One of the methods used effectively in the detection and classification of medical images is the U-Net architecture [46]. It takes the name “U-Net” from its shape. Its architecture includes first the encoder and then the decoder sections. The encoder, or contraction section, is divided into three. These are convolution, pooling, and deeper feature extraction layers. The decoder consists of upsampling, concentration, and convolution layers. There is also a bottleneck section formed at the end of the encoder and the beginning of the decoder. This part includes more convolution operations. When we look at its general definition, the U-Net architecture is used to extract low-resolution features from the input image. The output image is obtained by looking at these features. This architecture, which is mostly used in segmentation processes, is combined with other methods of deep learning. In this way, it becomes a technological approach that can be used in the detection of dental diseases. Considering its application in medical images and the scope of this study, eight studies—numbered S10, S14, S15, S18, S19, S22, S23, and S24—that employed U-Net were primarily included. In the S10 study [20], the objective was to detect apical lesions using CNNs. In the experiments, a dataset of 1000 panoramic images was divided into 80% for training, 10% for validation, and 10% for testing. In the study where U-Net architecture was used, lesion detection performed on the data was evaluated with different metrics. As a result of the segmentation process where Intersection over Union (IoU) thresholds were 0.3, 0.4, and 0.5, respectively, F1-scores were found to be 82.8%, 81.5%, and 74.2%, respectively. In study S14 [24], a deep learning-based method was proposed using CBCT images for the detection and segmentation of periapical lesions. A total of 61 periapical root images taken from 20 CBCT devices were used for the study. In the study where U-Net architecture was used as a lesion detection method, five classes were determined for segmentation. These classes are “lesion”, “bone”, “tooth structure”, “background”, and “restorative materials”, respectively. The accuracy of the U-Net architecture in which lesion detection was performed was found to be 93%. Similarly, different metrics and accuracy values were used for each class in the segmentation. Dice indices for each class gave 52%, 78%, 74%, 95%, and 58% results for lesion, bone, tooth structure, background, and restorative materials, respectively. When the results were examined, it was emphasized that the necessary conditions for automatic analysis can be provided by developing deep learning techniques on CBCT images used for lesion detection. Finally, the performance of deep learning-based U-Net architecture was evaluated for the detection of periapical lesions in the study S15 [25]. A total of 195 CBCT images were used as data, and these data were focused on the detection of small lesions. The grading of periapical lesions was based on the size of the lesions. Then, training was performed using a deep learning method. The results were evaluated according to sensitivity and specificity. U-Net architecture showed 86.7% sensitivity and 84.3% specificity in detecting periapical lesions according to their size. It was anticipated that these results will reach higher accuracy and reliability with the development of the algorithm in further studies. Apart from these, the remaining five studies generally presented an architecture that assists the methods in the effective detection of periapical lesions.*AlexNet:* The AlexNet CNN method, which has an important place in the use of artificial neural networks, basically has five convolutional and three fully connected layers [47]. In addition to fully connected layers, it provides effective results for the learning transfer method today. AlexNet forms a structure used in the formation of other networks, especially in object detection. Its structure includes various parts such as convolution, pooling, fully connected layers, and activation functions. There is a “dropout” function that helps prevent overfitting and enables its use in large datasets by optimizing GPU usage. In terms of primary studies, object detection was performed with AlexNet in three articles. In the study S8 [18], where the apical lesion was detected, the highest accuracy was achieved thanks to the use of AlexNet. With the AlexNet CNN model, 92.5% diagnostic accuracy was obtained on panoramic, periapical, and CBCT images. In the study S9 [19], where the periapical lesion was detected with AlexNet, the accuracy rate was 96.21%, and dentists were prevented from manually searching for the apical lesion, thus providing orientation to different dental diseases. In the study S27, Sajad et al. [48] performed the classification of periapical lesions located on the roots of teeth that are too small to be seen by dentists. This process was carried out in two stages. In the first stage, features were extracted using AlexNet and training was performed with a Support Vector Machine (SVM) and CNN. In the second stage, features were extracted from fully connected layers using the learning transfer method, and thus, lesion classification was performed based on the most meaningful features with the help of the softmax function without the need for data augmentation. As a result of these two stages, the data were augmented again then transferred to the SVM classifier, and 98% accuracy was achieved. As a result of the primary studies, it was predicted that AlexNet’s revolution can be an example for the field of health applications. 

*You Only Look Once (YOLOv3, YOLOv5, YOLOv8):* In terms of technological approaches, three methods that are added to the study together and meet different versions of YOLO object detection are used in the literature [49]. When their individual definitions are examined, YOLOv3 is a neural network created based on 53-layer DarkNet53 and consists of 106 layers in total. It has only one fully connected layer. Thanks to its deep structure, it can achieve high accuracy and perform object detection by boxing. YOLOv5 creates a trainable structure using PyTorch libraries. The system offers faster and lighter processing performance than YOLOv3. The YOLOv8 approach trains the data by integrating the PyTorch library into its structure like YOLOv5. It is based on YOLOv5 in terms of structure but increases training performance with improvement and optimization. In general, YOLOv8 can be suggested as a higher version of the YOLOv5 method [50]. There are five selected primary studies in which YOLO is used in the detection of dental diseases. The detection of different lesions in terms of content was performed with these approaches. In the S1 study [11], the detection was performed via YOLOv3 on periapical lesions verified by different experts. In the study, accuracy was 86.3%, specificity was 76%, sensitivity was 92.1%, the positive prediction was 86.4%, the negative prediction was 86.1% and finally the F1-score was 89%. In the study S4 [14], a machine learning-based support system was developed for the detection of dental diseases. The aim of the system was to detect dental conditions such as periapical lesions and missing teeth on panoramic images. For this purpose, data belonging to 733 patients were collected from Future University, Egypt. Using these data, a total of six dental diseases such as missing teeth and periapical lesions were determined using the YOLOv5 detection method. As a result of the detection of six classes, metric results of 0.61 mAP@0.5 and 0.28 mAP@[0.5–0.95] were obtained. As a result of the study, it was predicted that more diseases could be detected with the developing technology other than these six classes. In the study S12 [22], the aim was to detect apical lesions using panoramic images. The advantages and effectiveness of AI were also included during this detection process. In the study, where the dataset consisted of 306 panoramic images and 400 apical roots obtained from them, the F1-score, specificity, and sensitivity metrics took the values of 71%, 56%, and 98%, respectively. A comparison was made using these values, and it was determined that AI, through a YOLOv3-based detection system, had an assisting effect on dentists in making diagnoses. In the article numbered S7 [17], the performances of the YOLOv5 and YOLOv8 methods were compared on the dental lesion images. When the results were examined, it was stated that YOLOv8 was more effective in detecting lesions. 

*Convolutional Neural Network (CNN):* It is one of the deep learning methods that can be used especially in operations such as audio and image recognition [51]. Since it is an acronym for CNN, it is basically included in the structure of other networks. Since it is a building block for other networks, it also has activation functions, pooling, and fully connected layers along with the convolutional layer [52]. It learns the features in terms of the image using a hierarchical structure. Considering its structure and features, it is among the latest technological approaches that can be used in the detection of dental lesions. As the primary studies, articles numbered S5, S13, S19, and S29 were added to this systematic review since they used CNN to detect lesions. Among these, in the study S5 [15], as previously mentioned, the authors tried to increase the detection performance by using the ensemble structure with the method called Multi-Input Deep Convolutional Neural Network Ensemble (MI-DCNNE) in the detection of dental caries. In the primary study S13 [23], the effective use of AI to detect periapical lesions in panoramic radiographs was investigated since some lesions are too small to be seen visually. In this context, the authors marked 18618 periapical root areas in 713 panoramic images. Afterward, they classified the periapical lesion as present/absent and detected it with the help of CNN architecture. The average accuracy was 74.95%, with a sensitivity of 81% and specificity of 86%. As a result, it was seen that AI achieved successful results in an object type that is difficult to see such as periapical lesions. In the primary study S19 [29], the determination of periapical pathology was carried out for the detection of lesions and similar dental diseases in panoramic radiographs. The process performed is segmentation and the study compared the pathological findings obtained from here with the metric values obtained from deep learning methods in the detection of diseases. Two different deep learning methods were applied to the pathological findings extracted from 250 panoramic images. These are U-Net and Mask-RCNN. Both methods were used and compared with different metrics. These metrics are accuracy, precision, sensitivity, Dice index, and F1-score. For comparison, U-Net accuracy was found to be 98.1% and Mask-RCNN accuracy was found to be 46.7%. When the results were examined, it was seen that U-Net architecture performed better in lesion segmentation and detection. It was determined that these results can be improved when sufficient data are obtained. In another study, S29 [39], periapical lesions were detected using an X-Ray imaging technique. The situation that is emphasized here is the development of imaging techniques. It has been argued that new imaging techniques should be created for objects that are difficult to detect, such as periapical lesions, in addition to the X-Ray images used. As a result of the study, the detection accuracy of anomalies such as periapical lesions in teeth was found to be 95.85%. As the studies show, the CNN approach creates an auxiliary network structure similar to the U-Net architecture. 

*GoogleNet:* The GoogleNet deep learning model known as InceptionV1 was developed by Google [53]. The most notable aspect of this method, first introduced to the literature following its success in a competitive setting, is its utilization of “inception” blocks to perform the detection process. These blocks are formed by receiving filters of different sizes in parallel at the same time. Thanks to parallelism, different information can be received at the same time, and while the number of parameters decreases, performance is increased. Since accuracy is not achieved during these operations, the Rectified Linear Unit (ReLu) activation function is used [54]. Apart from these, it is thought that effective results can be obtained when used in dental lesion detection by reducing the number of parameters in the GoogleNet deep learning model with pooling layers [53]. Generally, the usage areas of this model are classification, object detection, and segmentation. In the primary studies S6 and S8, GoogleNet was utilized as a comparative model. In the study S6 [16], GoogleNet achieved an accuracy of 97.10%. In S8 [18], different networks were evaluated and compared. The aim was to identify the network that would best detect apical lesions. While the highest accuracy was achieved with AlexNet in the scope of the study, GoogleNet reached an accuracy of 89.36%. It is evident that GoogleNet, with its distinctive “inception” block structure, can be improved in the future and has the potential to achieve high accuracy in dental lesion detection. 

*Denti.Al:* A system based on AI named Denti.AI provides dentists with automatic information about pathologies in images. Denti.AI examines and detects using X-Ray images of the tooth. For this system-based deep learning model, study S11 was included in this systematic review as a primary study. In study S11 [21], the detection of apical lesion radiolucencies was performed on periapical images. The detection process, which was previously carried out on CBCT images, was conducted this time using 68 intraoral periapical images in a way that also highlighted the effectiveness of deep learning methods. In the two-part process conducted with the data added to the Denti.AI deep learning tool, different metric values were obtained for Reader 1 and Reader 2 and compared. As a result, the use of periapical radiographs in the detection of apical lesion radiolucencies increased the accuracy by 8.6% according to the alternative Free-Response Operating Characteristic (FROC) curve and Average Free-Response Operating Characteristic—Area Under Curve (AFROC-AUC) metrics. The study examining the effect of the periapical imaging technique on apical lesions meets all internal criteria. 

*Visual Geometry Group (VGG16, VGG19):* VGG16 and VGG19 are deep learning models used in different situations and consist of two networks with three fully connected layers, similar to AlexNet [55]. These networks have achieved significant success, were developed within the scope of competition, and have reached the present day. As a result, they have become methods that can be used in dental lesion detection. The structure of VGG16 consists of sixteen layers. Thirteen of these sixteen layers are convolutional layers, and three are fully connected layers. The convolutional layers use 3 × 3 filters and apply the ReLU activation function [56]. Additionally, maximum pooling is performed with 2 × 2 image sizes [57]. After training, operations such as object detection can be performed. Furthermore, in the transfer learning method, the three fully connected layer structures provide effective feature extraction. In the case of VGG19, the network consists of convolutional layers followed by three fully connected layers [58]. VGG19 is the more advanced version of VGG16, with a deeper network structure. However, the increase in depth brings with it an increase in cost and number of parameters [59]. VGG19 is a deep learning model that can be selected for segmentation, object recognition, and similar image processing tasks. In this systematic review study, two studies that performed lesion detection in this context are discussed. In the study S8 [18], AlexNet was used as the third comparison model for GoogleNet. Here, an accuracy of 87.94% was achieved. In the other primary study selected, S26 [36], two classes were defined, and the detection process was performed on these classes. The first class consisted of healthy teeth, and the second class consisted of teeth with endodontic lesions. Detection was performed using the DenseNet-121 network after the VGG16 automatic classification in combination with a Siamese network. The dataset consisted of 1000 sagittal and coronal slices extracted from 1000 CBCT images. The methods were tested, and in addition to achieving an acceptable classification performance, a detection accuracy of 70% was obtained. Based on this result, it was suggested that the Siamese network could be combined with different deep learning methods in the future to provide higher lesion detection rates. As observed in the primary studies, VGG16 and VGG19 constitute a viable structure for lesion detection. 

*DentaVN:* The DentaVN software, presented as a new approach in lesion detection, was used in the S28 study, and as a result of inclusion criteria, the S28 study was included in the primary studies. In the study S28, Ngoc et al. [38] state that the main purpose of their research is to provide evidence for the use of AI in disease diagnosis. In the study, which focused on periapical lesions, it was noted that there was limited research on this subject. For this purpose, machine learning-based software called DentaVN was developed using the Faster Region-based CNN (Faster R-CNN) architecture. This software used parameters of Faster-RCNN and detected periapical lesions with a 95.6% accuracy rate, which was also validated by dentists. The sensitivity and specificity metrics in the study were found to be 89.5% and 97.9%, respectively. Considering the results, it was concluded that the DentaVN software can serve as a supportive tool for dentists and is effective in the detection of periapical lesions. 

*RetinaNet:* The deep learning model RetinaNet provides the location information of objects in an image and uses it for classification [60]. It is primarily defined as a computer vision method. The problem at hand is to balance high accuracy with fast computation. In terms of its features, RetinaNet can be divided into three parts. The first part is the “focal loss” function, which is used to separate objects classified with different weights [61]. It treats examples as positive and negative, eliminating the negative examples from the system’s use, allowing the positive examples to stand out. In the second part, RetinaNet performs object detection in a single stage [62]. This enables the model to run quickly. It performs feature extraction by integrating ResNet or ResNeXt networks into itself [63]. This is an effective method, particularly in detecting abnormalities in medical images. In study S17 [27], the authors used the RetinaNet deep learning model for the detection of lesions in tooth roots, manually labeled from panoramic radiographs. In the study, periapical lesions were trained using ten different deep learning methods on 457 panoramic radiographs, and the results were evaluated using various metrics. The metrics included accuracy, sensitivity, precision, and F1-score, respectively. The data were divided into 80% training, 10% validation, and 10% test sets. In the study, accuracy ranged from 67.3% to 81.2%, sensitivity from 74% to 91%, precision from 82% to 93%, and finally, F1-score from 80% to 89.5%. The study, which yielded different results for each deep learning method, found that the best model was RetinaNet, and the best performance was achieved with Adaptive Training Sample Selection (ATSS). As a result, it was suggested that the RetinaNet method may be used effectively in clinical settings in the future with further method trials. 

*SqueezeNet:* It has been argued that the cost and number of parameters should be reduced in the field of deep learning [64]. In this context, the SqueezeNet deep learning model has become available with fewer parameters and faster performance. It aims to achieve high detection and classification accuracy by reducing memory usage. The layers of the SqueezeNet model are two: the “squeeze layer”, which reduces the depth of the input data using a 1 × 1 filter, and the “expand layer” to expand the data [65]. When compared, it contains 50 times fewer parameters after training compared to network structures such as Alexnet [66]. The advantage of this model compared to other models is that it reduces memory usage considerably with its compact structure and achieves high accuracy by including speed. It can detect objects from many embedded systems and medical images. In addition, since it does not have a fully connected layer, it is not a suitable network model for transfer learning approaches. In study S6 [16], this model was compared with GoogleNet. In this comparison, the SqueezeNet model performed better than GoogleNet. The accuracy value reached a high value of 99.9%, detecting lesions such as tooth decay. The SqueezeNet deep learning model shows that it is open to effective use in the future with this accuracy value. 

*Segment Anything Model (SAM):* This is a deep learning model developed by Meta’s Fundamental AI Research (FAIR) as a state-of-the-art instance segmentation model [67]. The model identifies each object in an image and assigns a mask to it. In the output image, each object is represented with distinct colors and patterns. While this model is effectively used for object identification, it demonstrates particularly good results in medical imaging. However, it has not yet achieved sufficient accuracy in lesion detection. In study S7 [17], this method was employed for comparison purposes but was found to be less effective, achieving only a 60% accuracy rate compared to other deep learning models. Therefore, it is considered open to improvement and is expected to yield better results in the future. 

*ResNet50:* ResNet50 is a deep learning model developed by Microsoft Research, consisting of 50 layers as its name suggests [68]. The model aims to achieve high accuracy in image recognition and object detection. In its 50-layer architecture, the initial layers extract features from the input image using a 7 × 7 filter. The “residual blocks” layer, which prevents gradient structure loss, forms convolution blocks in the order of 1 × 1, 3 × 3, and 1 × 1 [69]. Activation functions are present in the final layer of the ResNet50 deep learning model. Within the context of this systematic review, study S8 [18] compared ResNet50 with AlexNet, GoogleNet, and VGG19 deep learning models for the detection of apical lesions. In this comparison, the accuracy of the ResNet50 model was found to be 86.65%. Furthermore, ResNet50 serves as the foundation for networks such as RetinaNet. Based on this evaluation, the ResNet50 model is considered an improvable and effective method. 

*DetectNet:* DetectNet is a deep learning model developed by NVIDIA, with the primary purpose of object detection [70]. It offers an advanced structure for real-time and video object detection. The working system is training-based and designed for GPU usage. The model contains a CNN layer in its architecture. The “bounding box regressor” structure is employed to determine the exact positions of objects within the image [71]. In study S2 [12], the DetectNet deep learning model, which performs scaling and normalization on the input images as part of its pre-processing, detects cyst lesions on panoramic images. The cyst lesions were detected with an accuracy rate of 75–77%. While the model does not provide sufficient accuracy for healthcare applications, it is anticipated that it can be improved using different techniques. 

*MobileNetV2:* The MobileNetV2 deep learning model, considered by Google to provide both optimization and efficiency, is used as a solution to resource shortages in mobile and embedded systems [72,73]. It refers to the second generation of the MobileNet model. In the network architecture, inverse incremental convolutional connections are used instead of residual connections [74]. This creates a compact system with low computational requirements. The information obtained from the image is increased by expanding the first layer structure. Then, depth separable convolution is applied to the expanded features. In the activation function part, a linear transformation is applied over the features. In the S25 study [35], classification between radicular cysts and periapical lesions was performed using MobileNetV2, and YOLOv3 was later used for lesion and radicular cyst detection. Different metric values were obtained for each process in classification and detection. For radicular cyst classification, sensitivity was 95% and specificity was 86%, while for periapical lesions, these values were 77% and 93%, respectively. For detection, sensitivity was 83% for radicular cysts and 74% for periapical lesions. Based on these results, it can be stated that MobileNetV2, with its network structure and features, can be used in object detection and is a deep learning model that can be compared for lesion detection. Table 10 presents the state-of-the-art approaches in primary studies, along with their counts and percentages. 

#### 3.3.3. What Are the Data Augmentation Methods Used for Dental Images?

Insufficient data in medical images is a significant factor that can reduce the accuracy of deep learning tools. To address this limitation, pre-processing methods such as data augmentation are commonly employed. Data augmentation involves diversifying the dataset using various techniques, thereby enabling the development of more robust deep learning models [75]. Within the scope of this systematic review, the third research question examines the data augmentation methods utilized in the primary studies. The objective is to identify effective data augmentation techniques for the detection of dental diseases, a field particularly affected by data insufficiency. Twelve data augmentation techniques were identified from the primary studies, as outlined in Table 11. These include brightness and contrast adjustment, horizontal mirroring, trapezoid transformation, resizing, cropping, translation, scaling, shifting, sharpening, positioning zoom, and grayscale conversion. In some primary studies, these methods were applied in combination. Following the generation of augmented dental images, deep learning models were employed for tasks such as detection and classification, demonstrating the potential of these methods in improving model performance. 

*Brightness and contrast:* Brightness and contrast adjustments are techniques used to actively improve visual image quality [76]. Contrast is achieved through brightness adjustments. In the context of primary studies, brightness and contrast techniques were effectively utilized in four studies: S3, S4, S10, and S12. In study S3 [13], the need for sufficient data for the detection process was emphasized, with identified deficiencies in both the quantity of data and labeling accuracy. It was suggested that the detection process could be improved by employing diverse data augmentation and magnification methods. Similarly, in study S10 [20], brightness and contrast adjustments yielded high accuracy in detecting apical lesions. However, the lack of high-resolution images limited the study’s overall advancement. 

*Horizontal mirroring:* Horizontal mirroring can be defined as a data augmentation technique that involves flipping an image along its vertical axis, effectively swapping the left and right sides. This method was utilized in study S3 [13] to generate additional images following brightness and contrast adjustments. In the study, which focused on jawbone detection, the number of tooth images was increased through horizontal mirroring, and the detection process was conducted on the augmented dataset. 

*Trapezoid transformation:* The trapezoid transformation, a geometric technique that alters the perspective of an image to enhance visual diversity, was used in the S3 study. In the S3 study [13], where brightness, contrast, and horizontal mirroring were previously applied, trapezoid transformation was used in the final step to visualize the jawbone from alternative angles. This approach enhanced the system’s ability to increase accuracy in jawbone detection through deep learning. 

*Resize:* Resizing, a data augmentation method, adjusts image dimensions to facilitate the application of deep learning models for tasks such as detection or classification. In the S4 study [14], the resizing method was employed to detect dental diseases. This approach successfully generated sufficient data to identify different types of lesions. 

*Clipping:* The cropping technique involves trimming a specific part of an image to create a new one, aiming to extract different regions from the same data. In particular, focusing on the region that contains the detected lesion is crucial. The cropping technique was applied in two primary studies. In the S4 study [14], resizing was performed first, followed by cropping on the resized data, enabling the generation of additional data. Similarly, in the S25 study [35], cropping was applied to focus on specific parts of the tooth for enhanced analysis.

*Flip (horizontal flip, vertical flip):* The translation process is an image processing technique that involves flipping the image. It can be used to augment the dataset by horizontally and vertically flipping the images before adding them to the deep learning model [77]. A total of five primary studies employed the translation process, namely, S4, S10, S18, S23, and S24. In the S23 study [33], the authors performed deep learning-based detection using segmentation for apical lesion detection. Apical lesions were segmented from 470 panoramic images using the SpatialConfiguration-Net AI algorithm, which was developed based on the U-Net architecture. During this process, the images were flipped. A total of 63 apical lesions were detected across 47 panoramic images, which were used as test data. This approach led to the development of a deep convolutional neural network (D-CNN) algorithm for apical lesion detection. When evaluating the metrics, sensitivity was found to be 92%, precision 84%, and the F1-score 88%. After analyzing the results, the study questioned whether these outcomes were sufficient for use in dental clinics. In conclusion, the authors suggested that deep learning models could prove effective in detecting apical lesions. 

*Rescale:* Scaling is a data augmentation technique that modifies an image’s size to improve model accuracy. In the study S18 [28], when the scaling method was examined, periapical lesions were detected using a simple deep learning model. The goal of the study was to enhance panoramic radiograph images that contain periapical lesions. To achieve this enhancement, the illumination and contrast processes were performed in conjunction with scaling, using the Retinax algorithm. Afterward, the detection process was carried out using the U-Net architecture. Over 550 data points were used in the analysis of the modified images. According to the evaluation of the metric, the accuracy was 95.8%, the F1-score was 95.5%, and the sensitivity was 95.2%. The study demonstrated that this method could be effective and innovative for future applications in dental images. It was also suggested that these metric values could be further improved with additional data augmentation techniques. 

*Shift (width shift range, height shift range, blurry shift):* Shifting is a data augmentation technique that involves translating an image along the axes to enhance the generalization ability and performance of deep learning models. In the context of dental images, this technique aids in highlighting the region where the lesion is located, making it perceptible to both the model and the human observer. Shifting was applied in the primary studies S10 and S18 along both the horizontal and vertical axes. In the S10 study [20], this technique was used by shifting the axes to detect apical lesions. In the S18 study [28], shifting was performed in addition to scaling. Since scaling involves changes in pixel values, combining it with shifting allowed the model to achieve better results due to the enhancement provided by this additional data augmentation technique. 

*Rotation and reflection:* Rotating the image by a certain angle and reflecting it along a line are among the data augmentation techniques that contribute to the more effective execution of tasks such as detection and classification. Both methods were integrated and applied in five primary studies: S10, S24, S26, S27, and S29. In the S24 study [34], the authors highlighted that periapical lesions and associated conditions, such as tumors and cysts, are prevalent in dental diseases. A dataset was created with the help of 24 oral and maxillofacial specialists who identified dental diseases from 2902 panoramic images. Subsequently, deep learning methods based on U-Net architecture were tested using expert evaluations. The results showed a sensitivity of 92% and specificity of 84%. The study also reported an F1-score of 88%, a Dice coefficient of 88%, and an IoU metric value of 79%. With these results, 14 oral and maxillofacial experts confirmed that good results were achieved in lesion detection in panoramic images. The study concluded that the future development of deep learning in dental clinics is promising based on these metrics. However, it was noted that 49% of periapical radiolucencies were missed.

*Sharpening:* The sharpening technique, used to enhance details in an image, may assist in lesion detection by sharpening tooth contours, offering an effective approach in dental imaging. In the primary study S10 [20], sharpening was applied to panoramic radiographs to improve the detection of apical lesions. 

*Zoom:* Zooming, a technique that adjusts the scale of an image to enhance the model’s ability to learn from different object sizes, allows for the enlargement of the visible part of a lesion, enabling the deep learning model to better learn the regions with a higher likelihood of lesion occurrence. In the primary study S26 [36], a new dataset was generated by zooming in on the root regions of teeth in panoramic radiographs to detect periapical lesions. By narrowing the focus to potential lesion areas instead of the entire image, the study achieved a more targeted analysis. 

*Grayscale:* The grayscale filter simplifies image processing by converting Red, Green, Blue (RGB)-based images into grayscale, reducing the complexity in deep learning models and facilitating the detection of conditions such as lesions, cysts, and missing teeth. In the primary studies S27 [37] and S29 [39], this technique was utilized as a foundational pre-processing step for these purposes.

#### 3.3.4. What Are the Challenges and Proposed Solutions in Dental Lesion Detection?

In addition to the answers given to the questions in the articles examined within the scope of this systematic review, the determination of the difficulties in the studies and the subsequent solutions made for these difficulties were also included. Among the 29 studies determined as primary studies, six difficulties were determined to be found in common. These challenges are briefly explained below: 

*Lack of data:* Concepts such as AI, deep learning, and machine learning require sufficient data to work effectively. Researchers cannot adopt deep learning models due to the lack of data and the lack of explainability of the trained models [78]. Therefore, lack of data is seen as one of the difficulties that arise in the performance of the model used. This situation, which has a significant effect on the detection capability of the model, can reduce the accuracy obtained.

*Image quality:* Especially in the detection process from radiographic images such as panoramic and CBCT, the resolution must be of high quality. Image quality may deteriorate due to distortions such as blurring or the presence of noise. The distortion may cause a deficient, inappropriate image to be obtained [79]. As a result of this difficulty in medical images, the model cannot perform adequately. Therefore, image processing techniques such as noise reduction and cropping should be applied.

*Ability to generalize:* The generalization capability of a model is defined as the determination of how it responds to new data given to the system outside the dataset used. This problem can be solved by using the information obtained from the uncertain region of the feature space as test data. The parts of the model that are not seen in the training data give the ability to generalize to the test data [80]. 

*Lesion indistinctness:* In dental images, the lesion is a structure that is very difficult to see with the naked eye. For this reason, it is difficult to detect diseases such as cysts and caries in images with lesion ambiguity, and a solution is needed. Many deep learning models designed to detect lesions in medical imaging depend on AI systems. These systems can search for abnormally colored lumps with a specific shape. The parts of the system that can be fine-tuned can be healthy tissue colors or the minimum length and width range for a potential lump. Improvement of the system may prevent lesion ambiguity [81]. 

*Model complexity:* Complex deep learning models require more computation. Overfitting occurs as the complexity of the selected model increases. Model complexity is still in its infant stage in terms of deep learning [82].

*Risk of overfitting:* Although overfitting during training achieves high accuracy, these results are seen as incorrect. Overfitting refers to the situation where the network learns the features on the training dataset perfectly but does not generalize quite well on the test dataset [83]. This adaptation is a negative situation because it reduces the generalization ability of the model. 

After identifying the challenges, we found 13 proposed solutions to address these difficulties. It was determined that the proposed solutions varied depending on the challenges and the model used in the study. As a result, there were different suggestions for the same challenge. The abbreviations of the proposed solutions obtained from the studies are provided in Table 12 for reference.

The identified challenges and the corresponding proposed solutions are presented in Table 13. This table provides the abbreviations for the solutions addressing the difficulties encountered in the primary studies. For instance, in study S1, data augmentation, image pre-processing techniques, and model optimization were suggested to address the challenges of insufficient data, poor image quality, and limited generalization ability, respectively. Thus, the solutions to these challenges in study S1 are represented by the abbreviations PS1, PS2, and PS3.

## 4. Discussion

AI is increasingly utilized across various sectors such as healthcare, finance, education, transportation, and logistics, driven by technological advancements in today’s world. In the healthcare sector, particularly in areas like medical imaging and disease diagnosis, the results produced by deep learning models offer significant advantages in terms of workload reduction, accuracy, and speed. Specifically, in the context of dental lesion detection, which is the primary focus of this study, deep learning algorithms provide substantial benefits in image analysis, diagnosis, early detection, and treatment processes due to their high accuracy and reliability. This systematic review addresses four main questions and provides a comprehensive evaluation of the existing literature.

*1.* 
*Which lesion types were identified for detection and classification in dental panoramic, periapical, and CBCT images?*


Based on the obtained responses, five main types of lesions were identified: periapical lesions (62.07%), apical lesions (34.48%), cyst lesions (6.9%), dental caries (6.9%), and maxillary bone lesions (3.45%). The high prevalence of periapical lesions suggests that this type of lesion is common and often requires treatment, whereas the lower proportion of maxillary bone lesions indicates that these lesions are less frequent.

*2.* 
*What are the state-of-the-art approaches used to detect lesions in dental images?*


Among the 14 different deep learning models analyzed in this study, the most widely used model is U-Net, preferred by 27.59%. This suggests that U-Net is an effective and popular choice. Among the other CNNs, AlexNet, YOLOv3, YOLOv5, and YOLOv8 also appear to deliver effective results. YOLOv8 outperforms YOLOv5 in terms of speed and accuracy, indicating that this model offers distinct advantages in the application context. Additionally, models such as CNN, GoogleNet, Denti.AI, DentaVN, and RetinaNet stand out for their high accuracy rates and clinical applicability. These findings highlight the effectiveness of deep learning-based methods in dental applications and the criteria that should be considered when selecting a model.

*3.* 
*Which techniques are used for data augmentation in dental images?*


Twelve different data augmentation techniques, including brightness and contrast adjustments, horizontal mirroring, rotation, sharpening, positioning, zooming, and grayscale conversion, were identified as solutions to address data insufficiency. Among these, the flip and rotation and reflection methods were particularly significant in enhancing dataset diversity, each with a rate of 17.24%. These methods were the most commonly used. These findings highlight the effectiveness of data augmentation in improving model performance when working with limited data. However, different operations performed on images in data augmentation methods have certain limitations, primarily to maintain realism. In medical images, this can lead to anatomical misinterpretations. For instance, the vertical and horizontal rotation of X-Ray images can distort the image, causing misinterpretation due to changes in angle. In dental imaging, similar issues may arise due to misalignments between the upper and lower parts of the tooth. Therefore, data augmentation techniques must ensure realistic modifications based on the image type and domain. Misuse of these techniques may lead to erroneous results.

*4.* 
*What are the challenges and proposed solutions for detecting dental lesions?*


The primary challenges identified include data scarcity, image quality, generalization capability, lesion uncertainty, model complexity, and the risk of overfitting. These challenges present significant obstacles to the development of deep learning-based applications. To address these issues, thirteen solutions have been proposed. These include data augmentation, image pre-processing techniques, model optimization, model training, additional loss functions, cross validation, transfer learning, performance evaluation, model customization, data diversification, multi-model approaches, multi-scale CNNs, and expert opinion. These recommendations provide various strategic approaches to overcome challenges and enhance model performance. Addressing these challenges can make deep learning-based dental lesion detection systems more effective and reliable.

When examining the studies in this field, it becomes evident that while these studies primarily focused on evaluating the accuracy and performance of deep learning methods, our study offers a broader perspective on deep learning applications. This study reveals that periapical radiographs are the most commonly used imaging technique as input data for AI-based applications. Another key finding is that the frequent use of data augmentation methods enhances the generalization ability of the model and mitigates the issue of data insufficiency. The main challenges faced in this field, such as data scarcity, image quality, model complexity, and the risk of overfitting, along with potential solutions to address these challenges, are outlined in this study. It can be inferred from the literature that most systematic reviews on dental lesion detection have primarily concentrated on traditional machine learning algorithms. However, this study aims to fill significant gaps in the literature by thoroughly examining the effectiveness of deep learning algorithms and providing a comprehensive perspective.

### 4.1. Related Work

When the related literature is examined, it is seen that there are studies on lesion detection using different imaging techniques and deep learning models in this field. The diversity in the results of different imaging techniques and deep learning models used in lesion detection necessitates a systematic review of the existing literature in this field. There are various systematic review studies on lesion detection in dental panoramic images. Sadr et al. [84] investigated deep learning methods in the detection of periapical radiolucent lesions and divided the study into three categories: objective, systematic review, and meta-analysis. The aim of the authors was to conduct a systematic review and analyze the accuracy of detecting periapical radiolucent lesions across various image types. The accuracy of these detections was subsequently validated by experts. The meta-analysis utilized different image types, including posteroanterior (PA) radiographs, panoramic radiographs, and CBCT images. Classification, segmentation, and object detection on the images were performed using various deep learning methods. A total of 932 studies were analyzed for systematic review. Sensitivity, specificity, positive-similarity ratio, negative-similarity ratio, and diagnostic likelihood ratio were used as metrics in the analyses. These metrics were compared by experts to analyze the accuracy of detecting periapical radiolucent lesions. The deep learning tasks were then categorized into studies on the classification, segmentation, or detection of object periapical radiolucencies. Diagnostic performance and accuracy prediction values were calculated in the study. Liu et al. [85] increased the number of studies examined for meta-analysis to seven, including VGG16 and ResNet-18, which they used with some modifications. Thus, seven deep learning methods for segmentation, classification, and object detection were investigated. The average metric values of the included studies were calculated as sensitivity, specificity, positive-likelihood ratio, negative-likelihood ratio, and diagnostic rate. According to the meta-analysis, the included studies were determined to be highly reliable sources within the scope of GRADE (Grading of Recommendations Assessment, Development, and Evaluation). As a result of the study, deep learning methods that can be used based on the research of periapical radiolucent lesions in the literature were identified. Abusalim et al. [3] analyzed deep learning techniques for dental informatics in a systematic review study. In the study [3], the authors emphasize the importance of identifying effective methods for acquiring dental data, ensuring its accuracy, and utilizing it reliably. They argue that deep learning represents a promising technological approach in this data acquisition process, supporting their claims with evidence and insights from previous research on the topic. Our aim was to provide a general point of view for future researchers on this subject by conducting a systematic review through four research questions that have not been previously investigated in the literature on lesion detection in dental images using deep neural networks. Based on the answers to the questions, we determined that deep learning technologies have the potential to be a valuable tool for oral health professionals. However, deep learning systems are not mature enough to completely replace dental specialists; instead, they should be considered as additional tools to support dentists and specialists. This study is intended to be considered as an informative resource for researchers who will work on deep learning-based dental diagnosis using various medical images. This study aims to fill important gaps in the literature by providing a comprehensive review of deep learning methods for lesion detection in dental images and to demonstrate the effectiveness and challenges of these methods through a systematic review.

### 4.2. Limitations and Potential Threats to Validity

The scope of this study is limited by the following parameters:Date: This study covers primary studies published from 2019 to August 2024.Literature type: This study includes studies published in peer-reviewed journals and conference/workshop/symposium proceedings. Secondary systematic review studies, gray literature, and research studies such as surveys were excluded from the primary study candidate pool.Although this study focused on lesion detection from dental images, the automatic search result eliminated lesion types other than teeth with EC-4.We investigated data augmentation techniques to address the problem of class imbalance due to the difficulty of acquiring and labeling dental images.We focused on deep learning models that can make fast and accurate detection in medical images in recent years instead of studies involving machine learning algorithms in the primary study selection process.

In this study, a systematic review and validity assessment were conducted by considering the potential validity threats suggested by Petersen et al. [86] and Wohlin et al. [6]. Petersen et al. [86] stated that validity threats are generally related to the identification of the candidate article pool, selection bias, data extraction, and data synthesis. The choice of search terms and limitations of search engines can result in an incomplete pool of candidate articles. To minimize these risks, we carefully defined search terms and queried six widely used online databases in the medical field to find relevant studies on dental lesion detection, including deep learning models, data augmentation techniques, and lesion detection challenges and solutions. The application of inclusion and exclusion criteria may pose a potential threat to researcher bias and thus to validity. To mitigate this risk, the authors created a list of criteria and conducted a joint vote. The first and last authors jointly applied the criteria for each candidate article and selected the studies they agreed on. The validity of the data extraction process is an important aspect that directly affects the results of this study. To ensure the accuracy of the extracted data, the authors created categories iteratively. They aimed to reduce the risk of researcher bias by mapping relevant data to specified groups. When an author was unsure about the data to be extracted, they shared the situation with the other authors so that all authors could solve the problem through the mutual exchange of ideas.

Along with the validity threats proposed by Peterson el al. [86], we handled four additional potential validity threats—internal validity, construct validity, conclusion validity, and external validity—based on a standardized checklist developed by Wohlin et al. [6].

In terms of *internal validity*, the biggest threat encountered in this systematic review study was that despite searches on dental lesions, studies automatically obtained from search engines and including various types of lesions were eliminated by exclusion criteria. Furthermore, although our search scope and terms were broad, we found that most of the selected primary studies that addressed the research questions we identified did not mention adverse outcomes. Therefore, in order to evaluate the negative aspects of the studies, we analyzed all studies in depth and tried to draw negative conclusions.

*Construct validity* is concerned with assessing the difficulties encountered during the data extraction phase [6]. To reduce extraction bias, studies that can make a significant contribution to the results of the review are selected and evaluated. Studies that provide the most appropriate answers to the research questions are identified as primary studies. After the data extraction process is completed, unnecessary details and irrelevant results are eliminated by creating a data extraction form. In order to minimize bias, all selected studies are evaluated repeatedly until the final data extraction model is reached. Each study is read several times until the research question for which it is relevant and the clues and conclusions drawn from these questions are unified and the necessary information is carefully extracted.

In the context of *conclusion validity*, the quality metrics of the primary studies, where we could find answers to the research questions, were determined by the joint assessment of the authors, and the quality scores of the studies were clearly presented. Both quantitative and qualitative data analyses were conducted to ensure that the systematic review study was conducted rigorously and reproducibly. In addition, a data extraction form was used to record the search criteria, study selection process, inclusion and exclusion criteria, and findings from the research questions, providing a dynamic infrastructure where new additions could be easily made.

In terms of *external validity*, the findings obtained from primary studies in response to the identified research questions and the table contents created for each research question can be applied to other imaging techniques and image types in the medical field. This study is considered to cover all deep learning models and data augmentation methods used in dental images. In future studies, hybrid deep learning models can be used in this field.

### 4.3. Conclusions

In this study, a systematic review was conducted for lesion detection in dental images. In this comprehensive systematic review, periapical lesions—characterized by their occurrence at the apex of tooth roots—pose significant challenges in detection. While deep learning methods have demonstrated effectiveness in identifying these lesions, further advancements are required to enhance their accuracy and reliability. Cyst lesions, defined as fluid-filled structures, require the application of deep learning methods and data augmentation for accurate detection. While jawbone lesions can be identified with high accuracy, deep learning techniques have also proven effective in detecting dental caries and apical lesions. Across the reviewed studies, fourteen distinct deep learning models were identified. Among the identified models, U-Net emerged as the most commonly utilized deep learning model for segmentation, with a prevalence rate of 27.59% among the primary studies. U-Net demonstrated particularly high accuracy in the detection of apical lesions. Additionally, CNNs, including AlexNet, YOLOv3, YOLOv5, and YOLOv8, also yielded effective results in lesion detection. Notably, YOLOv8 outperformed YOLOv5 in both speed and accuracy. Furthermore, models such as CNN and GoogleNet have also demonstrated success in detecting dental lesions. Specialized software tools, including Denti.AI and DentaVN, stand out for their high accuracy rates and suitability for clinical applications, highlighting their potential for integration into real-world practice. RetinaNet has gained prominence for its advantages in fast and accurate detection. However, each method exhibits distinct strengths and limitations concerning accuracy and performance. Additionally, twelve data augmentation techniques were applied to enhance the quality and diversity of dental images used in these studies. Techniques such as brightness, contrast, horizontal projection, rotation, and sharpening were used to increase the diversity of the datasets and improve the generalization capability of the models. Among the data augmentation techniques, flip, rotation, and reflection were the most frequently employed methods in the primary studies. These approaches effectively improved model performance in tasks such as lesion detection and classification by addressing the issue of data insufficiency. Furthermore, six key challenges in dental lesion detection were identified, along with thirteen proposed solutions to address these challenges. From an operational perspective, integrating advanced deep learning models into clinical workflows can streamline diagnostic processes by reducing the time required for lesion detection and improving diagnostic consistency. These models, supported by various data augmentation techniques, have the potential to reduce labor-intensive procedures in dentistry. Addressing challenges such as data integration and model generalization through interdisciplinary approaches can further strengthen the effectiveness of these technologies.

In conclusion, the findings of this review provide a foundation for future research in dental imaging. Automatic detection systems and enhanced data augmentation methods can lead to faster and more accurate diagnostic processes. Moreover, combining high-accuracy deep learning models with innovative algorithms can improve scalability and applicability in healthcare settings. Model performance improvements can indirectly contribute to more efficient workflows in clinical settings. Effective collaboration between technical experts and clinicians will be essential to realize these advances and improve resource utilization in clinical workflows.

## Figures and Tables

**Figure 1 diagnostics-14-02768-f001:**
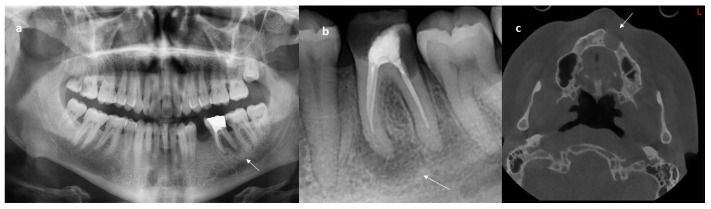
Radiographic methods used in radiology practice for dental lesion detection (white arrows indicate the lesion area in the relevant images): (**a**) a chronic apical periodontitis on panoramic radiography; (**b**) a chronic apical periodontitis on periapical radiography; (**c**) a radicular cyst on CBCT.

**Figure 2 diagnostics-14-02768-f002:**
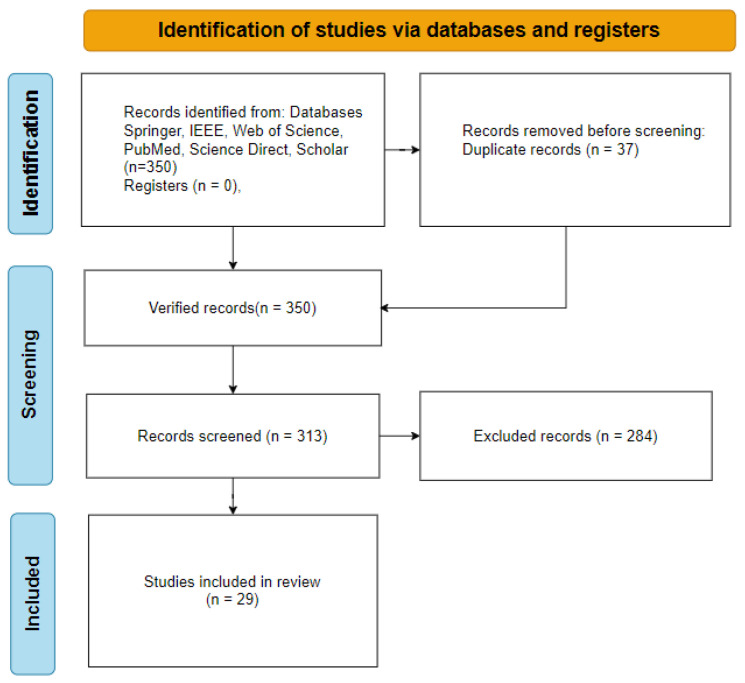
The PRISMA flowchart diagram of study selection process [4].

**Figure 3 diagnostics-14-02768-f003:**
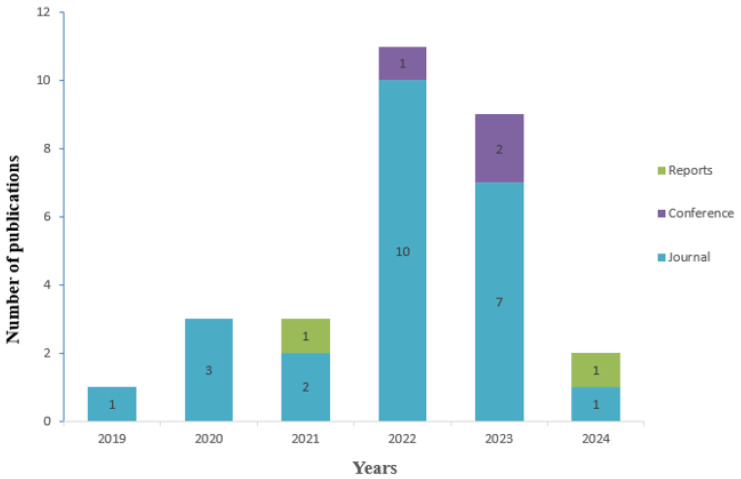
The number of primary studies according to publication type and publication year.

**Figure 4 diagnostics-14-02768-f004:**
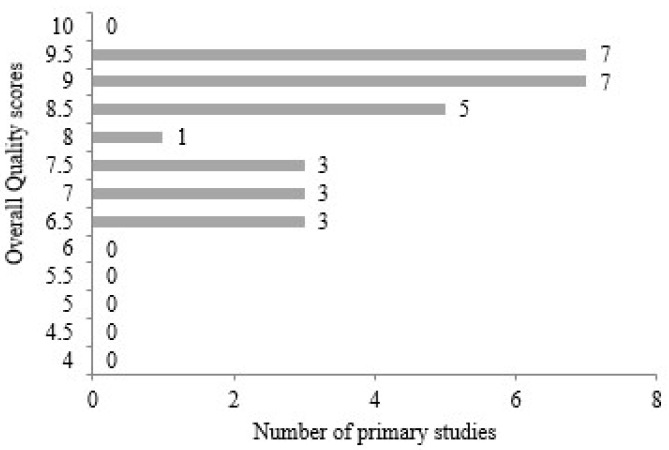
Overall quality scores of the primary studies.

**Figure 5 diagnostics-14-02768-f005:**
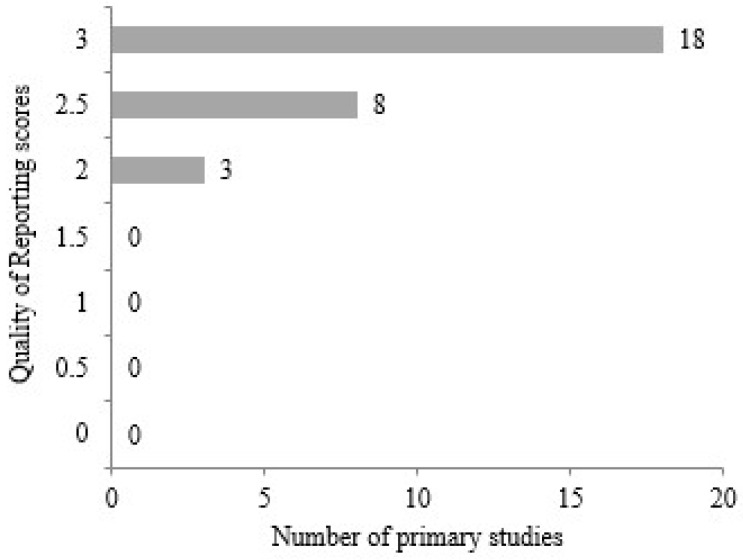
Reference reporting quality scores of the primary studies.

**Figure 6 diagnostics-14-02768-f006:**
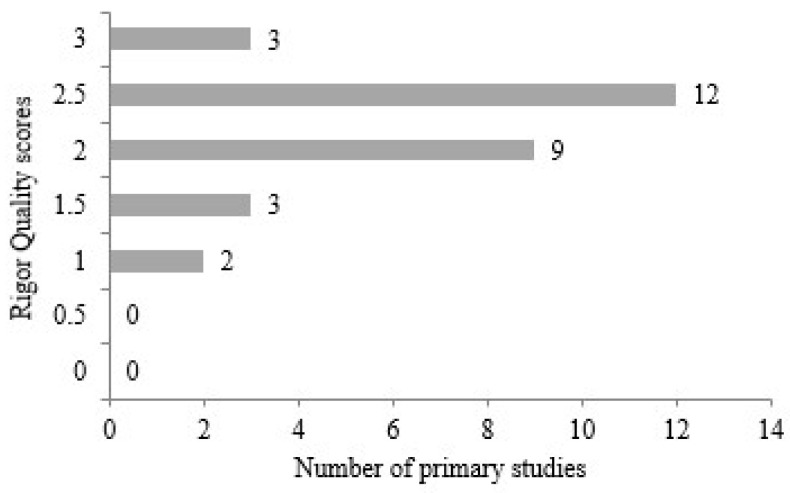
Rigor quality scores of the primary studies.

**Figure 7 diagnostics-14-02768-f007:**
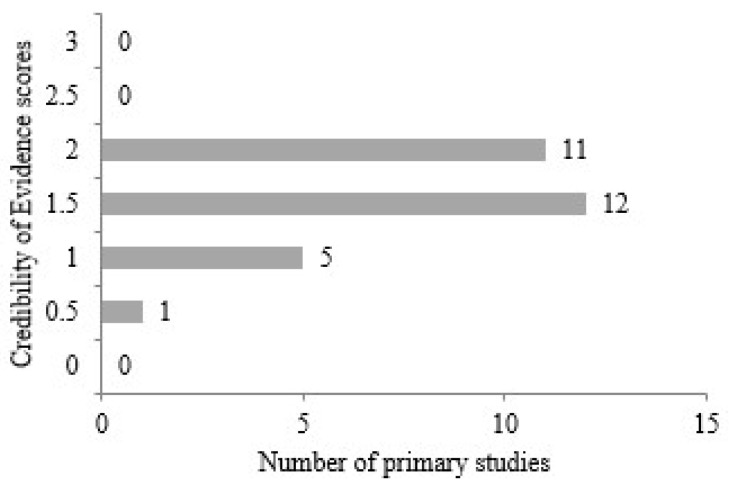
Credibility of evidence scores of the primary studies.

**Figure 8 diagnostics-14-02768-f008:**
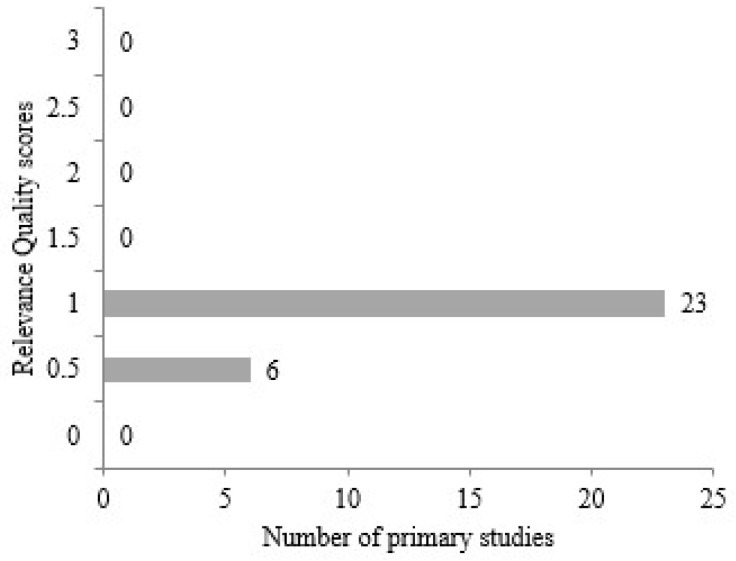
Relevance quality scores of the primary studies.

**Table 1 diagnostics-14-02768-t001:** Search queries.

Keywords	
**Q-1**	lesion detection OR object types AND dental images
**Q-2**	lesion detection AND deep learning methods
**Q-3**	state-of-the-art solutions AND dental images
**Q-4**	deep AND/OR machine learning methods in dental lesion detection
**Q-5**	application areas AND lesion detection

**Table 2 diagnostics-14-02768-t002:** Inclusion and exclusion criteria.

Exclusion Criteria	
**EC-1**	Studies that do not include state-of-the-art methods for lesion detection.
**EC-2**	Papers that do not have full text.
**EC-3**	Articles that do not fully address and discuss lesion detection.
**EC-4**	Articles that are systematic review articles, secondary studies, or surveys.
**EC-5**	Studies that only focus on lesion detection but do not include lesions in teeth.
**Inclusion Criteria**	
**IC-1**	Title or abstract/keywords include key terms.
**IC-2**	The summary of the study shows that the work is related to deep/machine learning methods.
**IC-3**	The language of the study is English or Turkish.
**IC-4**	The study detects lesions through panoramic/periapical/CBCT images.

**Table 3 diagnostics-14-02768-t003:** Database sources and number of selected studies.

Database	Total	Selected
Springer	126	3
IEEE	44	4
WoS	24	9
PubMed	44	1
ScienceDirect	24	5
Google Scholar	88	7
**Total**	**350**	**29**

**Table 4 diagnostics-14-02768-t004:** Primary studies extracted from the study selection process.

Paper Number	Authors	Title	Year	Source
S1	Moidu et al. [11]	Deep learning for categorization of endodontic lesion based on radiographic periapical index scoring system	2022	*Clinical Oral Investigations*
S2	Watanabe et al. [12]	Deep learning object detection of maxillary cyst-like lesions on panoramic radiographs: preliminary study	2020	*Oral Radiology*
S3	Gwak et al. [13]	Attention-guided jawbone lesion diagnosis in panoramic radiography using minimal labeling effort	2024	*Nature*
S4	El Bagoury et al. [14]	Dental Disease Detection based on CNN for Panoramic Dental Radiographs	2023	2023 IEEE Eleventh International Conference on Intelligent Computing and Information Systems
S5	Kaarthik et al. [15]	Detection and Classification of Dental Defect using CNN	2022	Proceedings of the Sixth International Conference on Intelligent Computing and Control Systems
S6	Chen et al. [16]	Missing Teeth and Restoration Detection Using Dental Panoramic Radiography Based on Transfer Learning With CNNs	2022	*IEEE Access*
S7	Demir et al. [17]	Deep Learning Based Lesion Detection on Dental Panoramic Radiographs	2023	Innovations in Intelligent Systems and Applications Conference
S8	Li et al. [18]	Detection of Dental Apical Lesions Using CNNs on Periapical Radiograph	2021	*Sensors*
S9	Chuo et al. [19]	A High-Accuracy Detection System: Based on Transfer Learning for Apical Lesions on Periapical Radiograph	2022	*Bioengineering*
S10	Song et al. [20]	Deep learning-based apical lesion segmentation from panoramic radiographs	2022	*Imaging Science in Dentistry*
S11	Hamdan et al. [21]	The effect of a deep-learning tool on dentists’ performances in detecting apical radiolucencies on periapical radiographs	2022	*Dentomaxillofacial Radiology*
S12	İçöz et al. [22]	Evaluation of an Artificial Intelligence System for the Diagnosis of Apical Periodontitis on Digital Panoramic Images	2023	*Nigerian Journal of Clinical Practice*
S13	Ba Hattab et al. [23]	Detection of Periapical Lesions on Panoramic Radiographs Using Deep Learning	2023	*Clinical Oral Investigations*
S14	Setzer et al. [24]	Artificial Intelligence for the Computer-aided Detection of Periapical Lesions in Cone-beam Computed Tomographic Images	2020	*Journal of Endodontics*
S15	Hadzic et al. [25]	Evaluating a Periapical Lesion Detection CNN on a Clinically Representative CBCT Dataset—A Validation Study	2024	*Journal of Clinical Medicine*
S16	Krois et al. [26]	Generalizability of deep learning models for dental image analysis	2021	*Nature*
S17	Çelik et al. [27]	The role of deep learning for periapical lesion detection on panoramic radiographs	2023	*Dentomaxillofacial Radiology*
S18	Latke et al. [28]	Detection of dental periapical lesions using retinex based image enhancement and lightweight deep learning model	2024	*Image and Vision Computing*
S19	Adnan et al. [29]	Multi-model Deep Learning approach for segmentation of teeth and periapical lesions on Pantomographs	2023	*Oral Surgery, Oral Medicine, Oral Pathology and Oral Radiology*
S20	Al-Awasi et al. [30]	Apical status and prevalence of endodontic treated teeth among Saudi adults in Eastern province: A prospective radiographic evaluation	2022	*King Saud University Saudi Dental Journal*
S21	Ekert et al. [31]	Deep Learning for the Radiographic Detection of Apical Lesions	2019	*Journal of Endodontics*
S22	Kirnbauer et al. [32]	Automatic Detection of Periapical Osteolytic Lesions on Cone-beam Computed Tomography Using Deep Convolutional Neural Networks	2022	*Journal of Endodontics*
S23	Bayrakdar et al. [33]	A U-Net Approach to Apical Lesion Segmentation on Panoramic Radiographs	2022	*Hindawi BioMed Research International*
S24	Endres et al. [34]	Development of a Deep Learning Algorithm for Periapical Disease Detection in Dental Radiographs	2020	*Diagnostics*
S25	Ver Berne et al. [35]	A deep learning approach for radiological detection and classification of radicular cysts and periapical granulomas	2023	*Journal of Dentistry*
S26	Calazans et al. [36]	Automatic Classification System for Periapical Lesions in Cone-Beam Computed Tomography	2022	*Sensors*
S27	Sajad et al. [37]	Automatic Lesion Detection in Periapical X-Rays	2022	Proc. of the 1st International Conference on Electrical, Communication and Computer Engineering
S28	Ngoc et al. [38]	Periapical Lesion Diagnosis Support System Based on X-Ray Images Using Machine Learning Technique	2021	*World Journal of Dentistry*
S29	Latke et al. [39]	A New Approach towards Detection of Periapical Lesions using Artificial Intelligence	2023	*Grenze International Journal of Engineering and Technology*

**Table 5 diagnostics-14-02768-t005:** Publication channels of primary studies.

No.	Publication	Long Name	Type	Instances
1	-	*Clinical Oral Investigations*	Journal	2
2	-	*Oral Radiology*	Journal	1
3	-	*Nature*	Reports	2
4	ICICIS	2023 IEEE Eleventh International Conference on Intelligent Computing and Information Systems	Conference	1
5	-	Proceedings of the Sixth International Conference on Intelligent Computing and Control Systems	Conference	1
6	-	*IEEE Access*	Journal	1
7	ASYU	Innovations in Intelligent Systems and Applications Conference	Conference	1
8	-	*Sensors*	Journal	2
9	-	*Bioengineering*	Journal	1
10	ISD	*Imaging Science in Dentistry*	Journal	1
11	-	*Dentomaxillofacial Radiology*	Journal	2
12	-	*Nigerian Journal of Clinical Practice*	Journal	1
13	JOE	*Journal of Endodontics*	Journal	3
14	J. Clin. Med	*Journal of Clinical Medicine*	Journal	1
15	-	*Image and Vision Computing*	Journal	1
16	-	*Oral Surgery, Oral Medicine, Oral Pathology and Oral Radiology*	Journal	1
17	-	*King Saud University Saudi Dental Journal*	Journal	1
18	-	*Hindawi BioMed Research International*	Journal	1
19	-	*Diagnostics*	Journal	1
20	-	*Journal of Dentistry*	Journal	1
21	ICECE	Proc. of the 1st International Conference on Electrical, Communication and Computer Engineering	Conference	1
22	-	*World Journal of Dentistry*	Journal	1
23	GRENZE	*Grenze International Journal of Engineering and Technology*	Journal	1

**Table 6 diagnostics-14-02768-t006:** Quality assessment criteria.

Quality Metrics	Question	Q. Type
Q1	Are the aims of the study clearly defined?	Reporting
Q2	Are the scope and the context of the study clearly stated?	Reporting
Q3	Is the proposed solution clearly explained and validated by an empirical study?	Reporting
Q4	Are the variables used in the study likely to be valid and reliable?	Relevance
Q5	Is the research process documented adequately?	Relevance
Q6	Are all study questions answered?	Relevance
Q7	Are the negative findings presented?	Rigor
Q8	Are the main findings stated clearly in terms of credibility, validity, and reliability?	Rigor
Q9	Do the conclusions relate to the aim of the purpose of the study?	Credibility
Q10	Does the report have implications in research and/or practice?	Credibility

**Table 7 diagnostics-14-02768-t007:** Quality scores of the primary studies.

Paper	REPORTING			RIGOR			CREDIBILITY		RELEVANCE		Rpr.	Rig.	Cre.	Rel.	Total
	**Q1**	**Q2**	**Q3**	**Q4**	**Q5**	**Q6**	**Q7**	**Q8**	**Q9**	**Q10**					
S1	1	1	1	1	1	1	0.5	1	1	1	3	3	1.5	2	9.5
S2	1	1	1	1	1	0.5	1	1	1	1	3	2.5	2	2	9.5
S3	1	1	1	1	1	0.5	0.5	1	1	1	3	2.5	1.5	2	9
S4	1	0.5	1	0.5	0.5	0.5	1	0.5	0.5	0.5	2.5	1.5	1.5	1	7
S5	1	1	1	0.5	1	0.5	0	0.5	1	1	3	2	0.5	2	7.5
S6	1	0.5	1	1	1	0.5	1	1	1	1	2.5	2.5	2	2	9
S7	1	1	1	1	1	0.5	0.5	1	1	1	3	2.5	1.5	2	9
S8	1	1	1	1	0.5	0.5	0.5	0.5	0.5	1	3	2	1	1.5	7.5
S9	1	0.5	1	1	1	0.5	0.5	1	1	1	2.5	2.5	1.5	2	8.5
S10	1	0.5	1	1	1	0.5	1	1	1	1	2.5	2.5	2	2	9
S11	1	1	1	1	0.5	0.5	0.5	1	1	1	3	2	1.5	2	8.5
S12	1	1	0.5	0	0.5	0.5	1	0.5	1	0.5	2.5	1	1.5	1.5	6.5
S13	1	1	1	0.5	1	0.5	1	0.5	1	1	3	2	1.5	2	8.5
S14	1	0.5	1	0.5	0	0.5	1	0.5	1	0.5	2.5	1	1.5	1.5	6.5
S15	1	1	1	1	0.5	0.5	1	1	1	1	3	2	2	2	9
S16	1	0.5	0.5	1	0	0.5	1	0.5	1	0.5	2	1.5	1.5	1.5	6.5
S17	1	1	1	1	1	0.5	1	1	1	1	3	2.5	2	2	9.5
S18	1	1	1	1	1	1	0	1	1	1	3	3	1	2	9
S19	1	1	1	0.5	1	0.5	1	1	1	1	3	2.5	2	2	9.5
S20	1	0.5	0.5	1	0.5	0.5	1	0.5	1	0.5	2	2	1.5	1.5	7
S21	1	0.5	0.5	0.5	0.5	0.5	1	0.5	1	1	2	1.5	1.5	2	7
S22	1	0.5	1	1	0.5	0.5	1	1	1	1	2.5	2	2	2	8.5
S23	1	1	1	1	1	0.5	1	1	1	1	3	2.5	2	2	9.5
S24	1	1	0.5	1	0.5	1	1	1	1	0.5	2.5	2.5	2	1.5	8.5
S25	1	1	1	1	1	1	0	1	1	1	3	3	1	2	9
S26	1	1	1	1	0.5	0.5	0	1	1	0.5	3	2	1	1.5	7.5
S27	1	1	1	1	1	0.5	1	1	1	1	3	2.5	2	2	9.5
S28	1	1	1	1	0.5	0.5	0.5	0.5	1	1	3	2	1	2	8
S29	1	1	1	1	0.5	1	1	1	1	1	3	2.5	2	2	9.5

**Table 8 diagnostics-14-02768-t008:** Object types retrieved from primary studies.

Types	Studies	Total	Percent
Periapical lesion	S1, S4, S9, S11, S13, S14, S15, S17, S18, S19, S20, S22, S24, S25, S26, S27, S28, S29	18	62.07%
Cyst lesion	S2, S25	2	6.9%
Jawbone lesion	S3	1	3.45%
Tooth decay lesion	S5, S6	2	6.9%
Apical lesion	S7, S8, S9, S10, S11, S12, S16, S19, S21, S23	10	34.48%

**Table 9 diagnostics-14-02768-t009:** Classification, segmentation and detection methods (YOLO: You Only Look Once, CNN: Convolutional Neural Network, VGG: Visual Geometry Group, SAM: Segment Anything Model).

AI Models and Methods	Classification	Segmentation	Detection
U-Net		X	X
AlexNet	X		X
YOLOv3, YOLOv5, YOLOv8			X
CNN	X	X	X
GoogleNet	X		X
Denti.Al			X
VGG16, VGG19			X
DentaVn			X
RetinaNet			X
SqueezeNet			X
SAM			X
ResNet50			X
DetectNet			X
MobileNetV2	X		

**Table 10 diagnostics-14-02768-t010:** State-of-the-art approaches in primary studies.

Approaches	Studies	Total	Percent
U-Net	S10, S14, S15, S18, S19, S22, S23, S24	8	27.59%
AlexNet	S8, S9, S27	3	10.34%
YOLOv3, YOLOv5, YOLOv8	S1, S4, S7, S12	4	13.8%
CNN	S5, S13, S19, S29	4	13.8%
GoogleNet	S6, S8	2	6.9%
Denti.Al	S11	1	3.45%
VGG16, VGG19	S8, S26	2	6.9%
DentaVn	S28	1	3.45%
RetinaNet	S17	1	3.45%
SqueezeNet	S6	1	3.45%
SAM	S7	1	3.45%
ResNet50	S8	1	3.45%
DetectNet	S2	1	3.45%
MobileNetV2	S25	1	3.45%

YOLO: You Only Look Once, CNN: Convolutional Neural Network, VGG: Visual Geometry Group, SAM: Segment Anything Model.

**Table 11 diagnostics-14-02768-t011:** Data augmentation methods for dental images in primary studies.

Methods	Studies	Total	Percent
Brightness and contrast	S3, S4, S10, S12	4	13.79%
Horizontal mirroring	S3	1	3.45%
Trapezoid transformation	S3	1	3.45%
Resize	S4	1	3.45%
Clipping	S4, S25	2	6.9%
Flip	S4, S10, S18, S23, S24	5	17.24%
Rescale	S18	1	3.45%
Shift	S10, S18	2	6.9%
Rotation and reflection	S10, S24, S26, S27, S29	5	17.24%
Sharpening	S10	1	3.45%
Zoom	S26	1	3.45%
Grayscale	S27, S29	2	6.9%

**Table 12 diagnostics-14-02768-t012:** Proposed solutions for the dental lesion detection using AI.

No	Proposed Solutions
PS1	Data augmentation
PS2	Image pre-processing techniques
PS3	Model optimization
PS4	Model training
PS5	Additional loss functions
PS6	Cross validation
PS7	Transfer learning
PS8	Performance evaluation
PS9	Model customization
PS10	Data diversification
PS11	Multiple model approach
PS12	Multi-scale CNN
PS13	Expert opinion

**Table 13 diagnostics-14-02768-t013:** Proposed solutions for the determined challenges.

No.	Challenges					
	Lack of Data	Image Quality	Ability to Generalize	Lesion Indistinctness	Model Complexity	Risk of Overfitting
S1	PS1	PS2	PS3	-	-	-
S2	PS1	PS2	-	PS4	-	-
S3	PS1	-	-	-	PS5	-
S4	PS1	-	-	PS6	-	-
S5	PS1	-	PS6	-	-	-
S6	PS1	-	-	-	PS7	-
S7	PS1	-	-	PS7	-	PS6
S8	PS1	-	PS6	PS7	-	-
S9	PS1	-	PS8	PS9	PS6	-
S10	PS1	PS2	-	-	PS8	-
S11	-	PS2	-	PS9	-	-
S12	-	PS2	-	PS9	-	-
S13	PS1	PS2	-	PS9	-	-
S14	PS1	PS2	-	-	-	-
S15	PS1	PS2	PS7	-	-	-
S16	PS1	PS2	-	-	-	PS10
S17	PS1	PS2	-	PS7	-	-
S18	PS1	PS2	-	-	-	-
S19	PS1	PS2	-	-	PS11	-
S20	PS1	PS2	-	-	-	-
S21	PS1	PS2	-	PS7	-	-
S22	PS1	PS2	PS7	PS12	-	-
S23	PS1	PS2	-	PS7	-	PS6
S24	PS1	PS2	PS6	PS12	-	-
S25	PS1	-	PS7	PS11	-	-
S26	PS1	PS2		PS11	-	
S27	-	PS2	-	PS13	-	-
S28	PS1	PS2	-	PS13	-	-
S29	PS1	PS2	PS7	PS9	-	-

## Data Availability

The original contributions presented in the study are included in the article/Appendix A, further inquiries can be directed to the corresponding author.

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
