# Peer review of "Comprehensive Insights into Artificial Intelligence for Dental Lesion Detection: A Systematic Review"

_diagnostics, 2024, doi:10.3390/diagnostics14232768_

Round 1
Reviewer 1 Report
Comments and Suggestions for Authors
The article summarizes dental lesion detection with AI-based models/architectures and possible solutions. The content and scope of the article provide a good overview of the literature.
The latest approaches to detect lesions in dental images, data augmentation methods, and the challenges and possible solutions for current AI-based dental lesion detection are detailed.
For completeness, I would first suggest that the architectures used for classification (such as Alexnet Vgg16) and segmentation (such as U-NET, Yolov5) be grouped under one heading.
When there is not enough data, the use of data augmentation techniques is important for AI. However, it should be discussed in the discussion section that not every data augmentation technique should be applied to the image. For example, rotating the image by a certain degree (0-10) is realistic. However, rotating the image at larger degrees is problematic. For example, no doctor rotates the image by 90 degrees or 270 degrees while examining it (for example, horizontal mirroring is not appropriate).
In which types of lesions is classification (or segmentation) used? I recommend that you add a summary table.
Author Response
MDPI DIAGNOSTICS
Special issue:Artificial Intelligence in Biomedical Diagnostics and Analysis
2024
One by One Answers to Reviewer Comments on the Article
Title: Comprehensive Insights into Artificial Intelligence for Dental Lesion
Detection: A Systematic Review
Manuscript id: diagnostics-3308188
Dear Editor,
The authors would like to thank the referees and the editor for their careful review of our manuscript “Manuscript ID: diagnostics-3308188” and for their comments and suggestions to improve the quality of the manuscript. We hope that the manuscript will meet the quality requirements for publication in MDPI Diagnostics.
The following Answers have been prepared to address the Comments of all reviewers individually. Each reviewer's comment is listed (in black) followed by our answers (in blue), with the word “Answer” at the end.
All changes have been added to the revised text. Sections revised according to reviewer comments are highlighted in yellow. However, the article has been revised from the beginning to the end in terms of academic writing and textual changes made outside of the referee comments are not highlighted to avoid confusion. This manuscript is extremely important to us. We have carried out all revisions with great care and meticulousness. We hope that the revised version will be accepted.
Editorial Office Comments
Please revise the manuscript found at the above link according to the reviewers' comments and upload the revised file within 10 days. Note the following check-list:
(I) Ensure all references are relevant to the content of the manuscript.
(II) Highlight any revisions to the manuscript, so editors and reviewers can see any changes made.
(III) Provide a cover letter to respond to the reviewers’ comments and explain, point by point, the details of the manuscript revisions.
(IV) If the reviewer(s) recommended references, critically analyze them to ensure that their inclusion would enhance your manuscript. If you believe these references are unnecessary, you should not include them.
(V) If you found it impossible to address certain comments in the review reports, include an explanation in your appeal.
We would like to draw your attention to the status of this invitation “Publish Author Biography on the webpage of the paper” -
https://susy.mdpi.com/user/manuscript/author_biography/81f5758e67715cab2fd859a1e1ffffaf.
If you decide to publish your biography, please remember to fill in it before your paper is accepted. If your manuscript requires improvement to the language and/or figures, you
may consider MDPI Author Services: https://www.mdpi.com/authors/english.
Please do not hesitate to contact us if you have any questions regarding the revision of your manuscript or if you need more time. We look forward to hearing from you soon.
Answer: Thank you for the editorial office's valuable comments and suggestions. The article was revised in line with all comments of the editor and referee. You can see all revisions below.
Reviewer-1’s Comments and Our Responses
The article summarizes dental lesion detection with AI-based models/architectures and possible solutions. The content and scope of the article provide a good overview of the literature. The latest approaches to detect lesions in dental images, data augmentation methods, and the challenges and possible solutions for current AI-based dental lesion detection are detailed.
We would like to thank the referee for all comments and contributions to improve the article. The comments have been processed item by item and revised in an annotated manner.
Comment 1: For completeness, I would first suggest that the architectures used for classification (such as Alexnet Vgg16) and segmentation (such as U-NET, Yolov5) be grouped under one heading.
Answer-1:
Thank you for this careful inspection. The segmentation, classification, and detection processes have been explained by grouping the architectures in the relevant section, as detailed on page 17, lines 367–381.
Comment 2: When there is not enough data, the use of data augmentation techniques is important for AI. However, it should be discussed in the discussion section that not every data augmentation technique should be applied to the image. For example, rotating the image by a certain degree (0-10) is realistic. However, rotating the image at larger degrees is problematic. For example, no doctor rotates the image by 90 degrees or 270 degrees while examining it (for example, horizontal mirroring is not appropriate).
Answer-2:
In response to the valuable feedback and suggestions provided by the reviewer, we have made the necessary revisions to the manuscript. Specifically, we clarified the data augmentation section and added the required paragraph on page 30, lines 870–877.
Comment 3: In which types of lesions is classification (or segmentation) used? I recommend that you add a summary table.
Answer-3: In accordance with the valuable feedback and suggestions of the reviewer, the necessary revisions have been implemented. Table 8 highlights the application of artificial intelligence models for classification, segmentation, and detection tasks on page 18.
Reviewer-2’s Comments and Our Responses
This is a systematic review about the use of deep learning models for dental lesion detection using dental panoramic, periapical and cone-beam computed tomography (CBCT) images. Given the rapidly expanding role of artificial intelligence (AI) in medicine and dentistry, this paper presents an interesting and contemporary topic, while it addresses current challenges and proposes solutions to improve AI-based methods for dental lesion detection. However, this paper presents significant issues that need to be resolved:
We would like to thank the referee for all comments and contributions to improve the article. The comments have been processed item by item and revised in an annotated manner.
Comment-1:
(x) The English could be improved to more clearly express the research.
|
|
Yes |
Can be improved |
Must be improved |
Not applicable |
|
Is the research design appropriate? |
( ) |
( ) |
(x) |
( ) |
|
Are the methods adequately described? |
( ) |
( ) |
(x) |
( ) |
|
Are the results clearly presented? |
( ) |
( ) |
(x) |
( ) |
Answer-1: Thank you for your valuable insights. To ensure the highest standards of English language quality, we carefully reviewed our manuscript. The paper was proofread by a professor who is an expert in the field, and additional grammar checks were conducted using tools such as Grammarly and ChatGPT.
Furthermore, we have refined the research design in accordance with your suggestions and enhanced the clarity of the methodological descriptions. Additionally, we have elaborated on the results to ensure greater clarity and comprehension.
Comment-2: The abstract in a manuscript should provide a precise and thorough overview of the entire study. Therefore, instead of simply mentioning “medical images” in the abstract, the authors should specify the exact types of input data, e.g., dental panoramic, periapical and cone-beam computed tomography (CBCT) images, used by the deep learning models for dental lesion detection.
Answer-2: Thank you for your careful inspection. The “medical images” phrase was removed then the abstract section was revised accordingly:
This study aims to provide a comprehensive systematic review on the use of deep learning methods for dental lesion detection in different types of imaging techniques such as panoramic, periapical and cone-beam computed tomography (CBCT).
Comment-3: Regarding the methodology, the authors mention that their systematic review follows the Preferred Reporting Items for Systematic Reviews and Meta-Analyses (PRISMA) guidelines. According to PRISMA guidelines, a systematic review should include the following sections: Title, Abstract, Introduction, Methods, Results, and Discussion. In this regard, I suggest the authors rename section 3 from “Systematic Literature Review” to “Methods”. Additionally, lines 115–119 and 172–179, which both describe the study selection process, should be combined and their refined form as well as the reference to the PRISMA flowchart diagram of the study selection process should be moved to the “Results” section as indicated by PRISMA guidelines.
Answer-3: In line with the valuable opinions and suggestions of the reviewer, the necessary corrections were made and the sections of our study were reorganized following the PRISMA guidelines. We sincerely thank the reviewer for their constructive suggestions, which have significantly enhanced the clarity and structure of our manuscript. In line with the reviewer's recommendations:
- Section Title Change: The title of Section 3 has been changed from "Systematic Literature Review" to "Methods" to better align with the PRISMA guidelines.
- Study Selection Process: In the initial version of the manuscript, the study selection process was described in the following sections (lines 115–119 and 172–179):
- The study selection process for Abusalim et al. [3] was discussed in lines 115–119,
- While our study's selection process was covered in lines 172–179.
To avoid confusion and clarify the distinction, these sections have been reorganized. - The study selection process for Abusalim et al. [3] now moved from the Related Work section to the Discussion section (page 30, lines 933–936).
- Our study's selection process has been moved to the "Results" section, in accordance with the PRISMA guidelines, along with the PRISMA flowchart (page 4-6).
These changes have improved the alignment with the guidelines and enhanced the overall organization of the manuscript.
Comment-4: Moreover, according to PRISMA guidelines, the authors should either provide registration information for their review protocol, including register name and registration number, or state that the review was not registered. Any financial or non-financial support for the review and any competing interests of review authors should also be declared.
Answer-4:
Thank you for pointing out the need to provide registration information in line with PRISMA guidelines. We would like to confirm that the review protocol has been registered with the following details and the registration is shown in the screenshot:
- Registry Name: Comprehensive Insights into Artificial Intelligence for Dental Lesion Detection: A Systematic Review [CRD42024607099]
- Registration Number: 607099
Furthermore, there is no financial or non-financial support for this review, and we declare that there are no competing interests among the authors. We added the “Conflict of Interest” section in the manuscript.
We added this sentence at the beginning of the Methods section: “We followed the PRISMA guidelines statement and registered on PROSPERO with the registration number CRD42024607099.”
Comment-5: Furthermore, the analysis of the study conducted by Sadr et al. in the “Related work” section should be more concise. Lastly, the “Related work” and “Limitations and Potential Threats to Validity” sections should be removed and their content should be incorporated into the discussion.
Answer-5: In line with the reviewer’s valuable comments and suggestions, the analysis of the study conducted by Sadr et al. (2023) in the “Related Work” section has been shortened (pages 30, lines 852–869). Additionally, the “Related Work” and “Limitations and Potential Threats to Validity” sections have been removed, and their content has been incorporated into the Discussion section (respectively, pages 30, lines 852–897, page 31 (related work) lines: 902-955, pages 32, lines 956 to page 33, lines 1020, page 33 (conclusion) lines: 1022-1050).
Comment-6: From a technical perspective, the manuscript also has some weaknesses:
- a) The use of abbreviations is not consistent throughout the manuscript. For example, the abbreviations “PRISMA” and “SLR” are introduced in the main body of the manuscript without being defined at their first use, and their definitions are only provided later in the text. The authors should clarify these abbreviations as “Preferred Reporting Items for Systematic Reviews and Meta-Analyses” and “Systematic Literature Review”, respectively, at first mention.
Answer-6: We sincerely thank the reviewer for pointing out the inconsistency in the use of abbreviations. In response to this comment, the abbreviations used in the manuscript have been clarified at their first occurrence, with "PRISMA defined as “Preferred Reporting Items for Systematic Reviews and Meta-Analyses”, "SLR" defined as “Systematic Literature Review”, "Artificial Intelligence" (AI), "Support Vector Machines" (SVM), and "K-Nearest Neighbors" (KNN) now fully explained.
Comment-7: b) Some sentences have poor syntax and are difficult to understand. For example, the sentences in lines 105–107: “In the study, first of all, what can be done to obtain the dental data to be investigated, determine its accuracy, and use it reliably.” and 397–398: “When we look at the article numbered S7, [18] aimed to detect lesions in the roots of teeth in their study.” need to be refined to improve readability.
Answer-7: We sincerely thank the reviewer for highlighting the issues with sentence structure and clarity. In response, we have revised the sentences identified in lines 105–107 and 397–398 to improve readability and clarity. Specifically:
- Line 105–107 has been rephrased as (It was moved to 933-936):
In the study by Abusalim et al. [2], the authors emphasize the importance of identifying effective methods for acquiring dental data, ensuring its accuracy, and utilizing it reliably. They argue that deep learning represents a promising technological approach in this data acquisition process, supporting their claims with evidence and insights from previous research on the topic.
- Line 397–398 has been rephrased as (The revised version was moved to 345-347):
In the primary study S7 [ 16 ], the authors aimed to detect lesions in tooth roots and the detection process was carried out using different deep learning methods on 660 images. After these processes, the results were compared and the method with the highest accuracy was decided
Comment-8: c) The authors should make footers below tables to define the abbreviations used in them.
Answer-8. Thanks for this careful inspection. We have added the necessary explanations for the abbreviations in Table 13 on page 29.
As authors of the manuscript, we would like to express our sincere thanks to the reviewers for their valuable comments. In the revised version of the manuscript, we have carefully focused on all of these valuable comments and hope that the revised version will satisfactorily meet academic standards.

Reviewer 2 Report
Comments and Suggestions for Authors
Comments and suggestions for authors are attached.

The language of the manuscript should be improved to enhance readability.
Author Response

(The authors gave the same response as above.)

Round 2
Reviewer 2 Report
Comments and Suggestions for Authors
My review repost is attached.

Author Response
MDPI DIAGNOSTICS
Special issue:Artificial Intelligence in Biomedical Diagnostics and Analysis
2024
One by One Answers to Reviewer Comments on the Article
Title: Comprehensive Insights into Artificial Intelligence for Dental Lesion
Detection: A Systematic Review
Manuscript id: diagnostics-3308188
Dear Editor,
The authors would like to thank the referees and the editor for their careful review of our manuscript “Manuscript ID: diagnostics-3308188” and for their comments and suggestions to improve the quality of the manuscript. We hope that the manuscript will meet the quality requirements for publication in MDPI Diagnostics.
The following Answers have been prepared to address the Comments of all reviewers individually. Each reviewer's comment is listed (in black) followed by our answers (in blue), with the word “Answer” at the end.
All changes have been added to the revised text. Sections revised according to reviewer comments are highlighted in red. However, the article has been revised from the beginning to the end in terms of academic writing and textual changes made outside of the referee comments are not highlighted to avoid confusion. This manuscript is extremely important to us. We have carried out all revisions with great care and meticulousness. We hope that the revised version will be accepted.
Editorial Office Comments
Please revise the manuscript found at the above link according to the reviewers' comments and upload the revised file within 5 days. Note the following check-list:
(I) Ensure all references are relevant to the content of the manuscript.
(II) Highlight any revisions to the manuscript, so editors and reviewers can see any changes made.
(III) Provide a cover letter to respond to the reviewers’ comments and explain, point by point, the details of the manuscript revisions.
(IV) If the reviewer(s) recommended references, critically analyze them to ensure that their inclusion would enhance your manuscript. If you believe these references are unnecessary, you should not include them.
(V) If you found it impossible to address certain comments in the review reports, include an explanation in your appeal.
If your manuscript requires improvement to the language and/or figures, you may consider MDPI Author Services: https://www.mdpi.com/authors/english. Please note the status of this invitation “Publish Author Biography on the webpage of the paper” -
https://susy.mdpi.com/user/manuscript/author_biography/81f5758e67715cab2fd859a1e1ffffaf.
If you wish to publish your biography, please complete it before your manuscript is accepted.
Please do not hesitate to contact us if you have any questions regarding the revision of your manuscript or if you need more time. We look forward to hearing from you soon.
Answer: Thank you for the editorial office's valuable comments and suggestions. The article was revised in line with all comments of the editor and referee. You can see all revisions below.
Reviewer-1’s Comments and Our Responses
The authors have successfully addressed most of my remarks and the revised manuscript has been improved. However, it still needs some refinement.
We would like to thank the referee for all comments and contributions to improve the article. The comments have been processed item by item and revised in an annotated manner.
Comment 1: The “Related work” section is currently presented as a single long paragraph and includes many details about the methodology of the cited studies. The authors should summarize the studies more concisely, focusing on their results. For instance, lines 940–946: “In total, six databases, namely Web Of Science, Scopus, ACM, Springer, Science Direct and IEEE, were searched. Each database was subjected to three stages. During the validation phase, 33 of the 1379 studies were removed as duplicates. In the screening, after excluding 1202 studies that were excluded, 22 studies that could not be obtained and 30 studies that were excluded after reading, 92 studies were included in the final stage. In the third part of the methodology, the studies were divided according to the questions” should be omitted
Answer-1:
The sentences on lines 940-946 of the text, which is mentioned in the relevant study section, were omitted.
Comment 2: In my previous review report, I suggested that the authors should include footers below the tables to define the abbreviations used in them. The authors added a footer below Table 13 in which they included the explanations for PS1–13, which is redundant as they have already cited these explanations in Table 12. Therefore, the footer below Table 13 is not needed. Instead, what the authors should do is add footers below any table that includes abbreviations, such as CNN, to define these abbreviations
Answer-2: The caption written as footer in Table 13 has been removed and all abbreviations were added to the related tables as footers. Besides, longer versions of abbreviations are also written where they first appear.
Comment 3: The “Conclusion” section should be condensed into a single paragraph and made more concise. The authors should not repeat the research questions as this adds unnecessary length to the section.
Answer-3: Repeated study questions have been removed. The final version of the Conclusion section is as follows
In this study, a systematic review was conducted for lesion detection in dental images. In the comprehensive systematic review, periapical lesions—characterized by their occurrence at the apex of tooth roots—pose significant challenges in detection. While deep learning methods have demonstrated effectiveness in identifying these lesions, further advancements are required to enhance their accuracy and reliability. Cystic lesions, defined as fluid-filled structures, require the application of deep learning methods and data augmentation for accurate detection. While jawbone lesions can be identified with high accuracy, deep learning techniques have also proven effective in detecting dental caries and apical lesions. Across the reviewed studies, fourteen distinct deep learning models were identified. Among the identified models, U-Net emerged as the most commonly utilized deep learning model for segmentation, with a prevalence rate of 27.59\% among the primary studies. U-Net demonstrated particularly high accuracy in the detection of apical lesions. Additionally, convolutional neural networks, including AlexNet, YOLOv3, YOLOv5, and YOLOv8, also yielded effective results in lesion detection. Notably, YOLOv8 outperformed YOLOv5 in both speed and accuracy. Furthermore, models such as CNN and GoogleNet have also demonstrated success in detecting dental lesions. Specialized software tools, including Denti.AI and DentaVN, stand out for their high accuracy rates and suitability for clinical applications, highlighting their potential for integration into real-world practice. RetinaNet has gained prominence for its advantages in fast and accurate detection. However, each method exhibits distinct strengths and limitations concerning accuracy and performance. Additionally, twelve data augmentation techniques were applied to enhance the quality and diversity of dental images used in these studies. Techniques such as brightness, contrast, horizontal projection, rotation and sharpening were used to increase the diversity of the datasets and improve the generalization capability of the model. Among the data augmentation techniques, flip, rotation and reflection were the most frequently employed methods in the primary studies. These approaches effectively improved model performance in tasks such as lesion detection and classification by addressing the issue of data insufficiency. Furthermore, six key challenges in dental lesion detection were identified, along with thirteen proposed solutions to address these challenges.
In conclusion, we hope that the findings will pave the way for further research. In the future studies, automatic detection systems and data augmentation methods using deep learning models may contribute to obtaining faster and more accurate results in dentistry. Moreover, the combination of different deep learning methods with high accuracy and the development of new algorithms may enable more effective detection of dental diseases
As authors of the manuscript, we would like to express our sincere thanks to the reviewers for their valuable comments. In the revised version of the manuscript, we have carefully focused on all of these valuable comments and hope that the revised version will satisfactorily meet academic standards.